# Towards Practical Reproduction of Stochastic Concept Bottleneck Models

## Abstract

Stochastic Concept Bottleneck Models (SCBMs) model dependencies among concept logits with a joint Gaussian distribution, enabling interventions on corrected concepts to propagate to related non-intervened concepts. We reproduce the main SCBM experiments on a synthetic correlated-concept dataset and two natural image datasets, comparing SCBM with Hard CBM, autoregressive CBM, and Concept Embedding Models. Our results broadly validate the original empirical findings: SCBM remains competitive in predictive accuracy, improves concept-probability calibration, and enables more efficient interventions, requiring fewer manual concept corrections to achieve comparable concept and target accuracy. Beyond empirical reproduction, we study the practical cost of reproducing SCBM. We identify implementation bottlenecks in the official codebase and introduce a substantially more efficient GPU-oriented reproduction framework. These changes reduce the practical reproduction cost to approximately 62 wall-clock hours on a single RTX 4090. Despite these implementation changes, the principal empirical findings remain largely stable, suggesting that the original conclusions do not depend on implementation-specific choices. However, once avoidable overhead is removed, the relative runtime behavior of the compared models changes substantially: SCBM's reported computational advantage largely disappears, indicating that its practical benefit comes not from raw efficiency but from exploiting concept dependencies during intervention. Overall, our study supports SCBM's core methodological contribution while clarifying its position among existing CBM variants.

## 1 Introduction

Concept Bottleneck Models (CBMs) provide an interpretable alternative to standard end-to-end neural networks by requiring predictions to be mediated through human-interpretable concepts (Koh et al., 2020; Havasi et al., 2022; Shin et al., 2023). Rather than explaining a prediction only after it has been made, CBMs expose an intermediate concept layer that users can inspect and, when necessary, correct. This test-time intervention mechanism is one of the central motivations for CBMs: if a predicted concept is wrong, replacing it with the correct value can improve the final target prediction. This makes CBMs attractive when interpretability is expected to support human oversight rather than only provide post-hoc explanations.

However, a key limitation of standard CBMs is that they typically treat concept predictions independently. As a result, correcting one concept changes only that concept and leaves the remaining concept predictions fixed, even when concepts are statistically dependent. This can make interventions inefficient when concepts are correlated. Stochastic Concept Bottleneck Models (SCBMs) (Vandenhirtz et al., 2024) address this limitation by learning a joint Gaussian distribution over the concept logits. The learned dependency structure allows interventions on corrected concepts to propagate to related non-intervened concepts. The original SCBM paper reports that this dependency-aware bottleneck improves concept-probability calibration and intervention efficiency while maintaining competitive predictive performance.

Our primary goal is to independently reproduce the main empirical claims of Vandenhirtz et al. (2024). Specifically, we examine whether learning concept correlations preserves predictive performance before intervention while improving concept-probability calibration; whether dependency-aware interventions that use

the learned concept correlations improve intervention effectiveness and efficiency in terms of both concept accuracy and target accuracy; and whether SCBM retains the training and inference efficiency of CBMs relative to other CBM variants.

Nonetheless, reproducing these results is practically nontrivial. As estimated by Vandenhirtz et al. (2024), reproducing the full set of experiments with the original implementation requires approximately 1,200 GPU hours on GeForce RTX 2080 hardware. Such a cost makes full reproduction difficult in practice for other researchers and comes at the expense of additional baselines, ablations, or robustness checks that could otherwise be run with the same compute. We used a more recent GeForce RTX 4090 for our reproduction, but the released implementation was still substantially slower than expected. This high reproduction cost motivated us to optimize the official implementation and report the resulting reduction in reproduction cost as a secondary part of our study.

To address these goals, we reproduce the experiments reported in the main text of Vandenhirtz et al. (2024) on a synthetic dataset and two natural image datasets. The natural-image experiments use the same datasets as the original paper, while the synthetic experiment follows the same data-generation procedure with adjusted hyperparameters, as detailed in Section 4.1 and Appendix B.1. Following the original comparison setup, we compare SCBM with Hard CBM (Havasi et al., 2022), autoregressive CBM (Havasi et al., 2022), and Concept Embedding Models (Espinosa Zarlenga et al., 2022), evaluating predictive performance, concept-probability calibration, and intervention performance.

In parallel, we refactor the official codebase and optimize the main computational bottlenecks in data loading, model execution, and intervention evaluation, including a GPU-accelerated solver for dependency-aware interventions. Besides reducing the practical cost of reproduction, these implementation changes also allow us to assess which conclusions of the original work remain robust under different implementation choices and which are more sensitive to the implementation.

**Contributions** Our study makes three contributions. (i) We broadly validate the main empirical claims of SCBM: SCBM remains competitive in predictive performance, improves concept-probability calibration, and enables more efficient interventions through dependency-aware concept propagation. (ii) We identify major implementation bottlenecks and introduce a GPU-oriented reproduction framework that substantially reduces the practical cost of reproducing SCBM. (iii) We revisit the runtime comparison reported in Vandenhirtz et al. (2024) and show that its reported computational advantage largely disappears once avoidable implementation overhead is removed, indicating that this efficiency claim is considerably more implementation-dependent than the other principal findings. This suggests that SCBM's practical benefit lies primarily in exploiting concept dependencies during intervention rather than in computational efficiency, providing clearer guidance on how SCBM should be positioned among existing CBM variants.

## 2    Related Work

CBMs (Koh et al., 2020) are typically trained on supervised data of the form $(\boldsymbol{x}, \boldsymbol{c}, y)$, where $\boldsymbol{x} \in \mathbb{R}^p$ denotes the input with $p$ covariates, $\boldsymbol{c} \in \mathbb{R}^C$ is a concept vector of $C$ human-interpretable concepts, and $y \in \mathbb{R}$ is the target label. A CBM factorizes prediction into a concept predictor and a target predictor, $\hat{\boldsymbol{c}} = h_\phi(\boldsymbol{x})$ and $\hat{y} = g_\psi(\hat{\boldsymbol{c}}) = g_\psi(h_\phi(\boldsymbol{x}))$, so that the final prediction is mediated by the predicted concept bottleneck.

The form of the bottleneck affects the faithfulness of the resulting concept-based explanation. In the soft CBM setting considered by Koh et al. (2020), continuous concept logits $\boldsymbol{\eta} \in \mathbb{R}^C$ are passed to the target predictor. Subsequent work argues that this can lead to information leakage: continuous logits may encode task-relevant information beyond the annotated concept values, allowing the target predictor to distinguish examples that should be indistinguishable from the concepts alone (Margeloiu et al., 2021; Mahinpei et al., 2021). Hard CBMs (Havasi et al., 2022) mitigate this problem by passing binarized concept values $\hat{\boldsymbol{c}} \in \{0,1\}^C$ to the target predictor, rather than continuous concept logits. Havasi et al. (2022) further use autoregressive concept predictors to capture correlations among concepts while retaining a hard bottleneck. In this model, each concept $c_i$ is predicted by a separate MLP that takes an intermediate representation from the encoder and the previous concepts $c_1, \ldots, c_{i-1}$ as input. This allows earlier concepts to influence later concept predictions, but makes the learned dependency structure dependent on the chosen concept ordering.

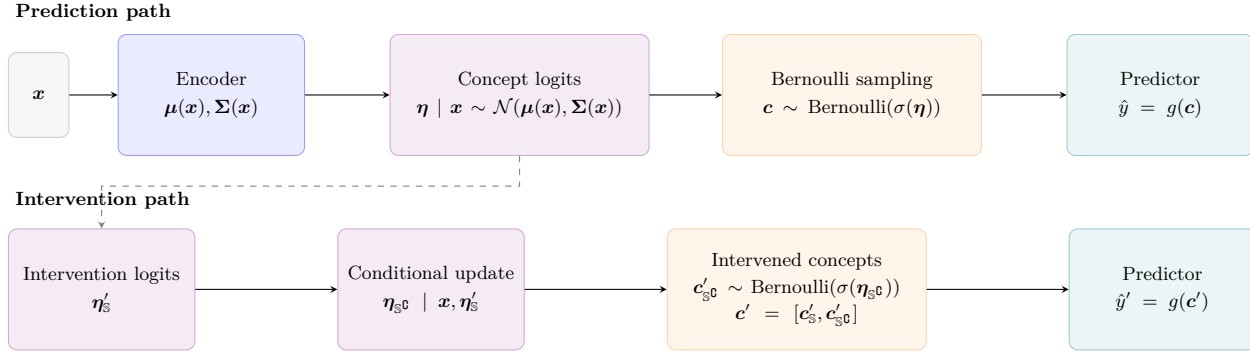

Figure 1: Overview of the SCBM framework. *Top:* During prediction, the encoder outputs a Gaussian distribution over concept logits $\boldsymbol{\eta}$, from which binary concepts $\boldsymbol{c}$ are sampled and passed to the predictor. *Bottom:* During intervention, for a set of intervened concepts $\mathbb{S}$, the corresponding concept logits $\boldsymbol{\eta}_\mathbb{S}$ are altered to the intervention logits $\boldsymbol{\eta}_\mathbb{S}'$, which are used to conditionally update the remaining logits $\boldsymbol{\eta}_{\mathbb{S}^\complement}$; the final prediction $\hat{y}'$ uses both the intervened true concepts and the updated non-intervened concepts.

Concept Embedding Models (CEMs) (Espinosa Zarlenga et al., 2022) take a different approach by representing each concept with a high-dimensional embedding rather than a single scalar. For each concept $i$, CEM learns positive and negative concept embeddings, denoted by $\boldsymbol{z}_i^+$ and $\boldsymbol{z}_i^-$. A learnable scoring function estimates the concept probability $\hat{p}_i$, and the concept representation is formed as $\hat{\boldsymbol{z}}_i = \hat{p}_i \boldsymbol{z}_i^+ + (1 - \hat{p}_i) \boldsymbol{z}_i^-$. The concatenation of all concept embeddings is then passed to the target predictor. CEMs therefore provide a strong embedding-based alternative to scalar concept bottlenecks, aiming to improve predictive performance while preserving concept-level interpretability. However, this expressiveness also creates room for information leakage: the embeddings may carry task-relevant information that is not captured by the binary concept values, potentially weakening the faithfulness of the concept-based explanation.

An attractive property of CBMs is that they allow users to replace a subset of predicted concepts with corrected values and then recompute the target prediction. An intervention policy determines the order in which concepts are intervened on. This order matters because concept interventions require human effort; correcting more informative concepts earlier can therefore improve performance with fewer manual interventions. The original CBM work considers random concept interventions (Koh et al., 2020). Later work studies intervention policies that select concepts according to some criteria such as predicted uncertainty, reducing the need for users to manually search over the full concept set (Sheth et al., 2022; Shin et al., 2023). However, these intervention policies primarily decide which concepts to correct; they do not by themselves update the remaining concept predictions after an intervention. This limitation is important when concepts are statistically dependent, because correcting one concept may provide evidence about other related concepts. Autoregressive CBM (Havasi et al., 2022) partially addresses this problem by conditioning later concept predictions on earlier concepts, so an intervention on an earlier concept can affect subsequent concept predictions. However, this dependency propagation is tied to a predefined concept ordering and the imposed order may not reflect a natural semantic or causal structure. SCBM (Vandenhirtz et al., 2024) addresses this gap by learning a joint distribution over concept logits, so that interventions can propagate information to correlated concepts through the learned dependency structure without imposing a fixed autoregressive ordering.

## 3 Method

### 3.1 Model Formulation

A schematic overview of the SCBM framework is shown in Figure 1. The SCBM framework models concept dependencies by replacing deterministic concept logits with an explicit joint distribution over latent concept logits. For each input $\boldsymbol{x}$, an encoder predicts the parameters of a Gaussian distribution over concept logits $\boldsymbol{\eta} \mid \boldsymbol{x} \sim \mathcal{N}\big(\boldsymbol{\mu}(\boldsymbol{x}), \boldsymbol{\Sigma}(\boldsymbol{x})\big)$, where $\boldsymbol{\eta} \in \mathbb{R}^C$, $\boldsymbol{\mu}(\boldsymbol{x}) \in \mathbb{R}^C$, and $\boldsymbol{\Sigma}(\boldsymbol{x}) \in \mathbb{R}^{C \times C}$.

Standard CBMs typically treat concept predictions as conditionally independent given the input. SCBMs relax this assumption, where conditioned on $\boldsymbol{\eta}$, each concept is modeled independently as a Bernoulli variable:

$$p(\boldsymbol{c} \mid \boldsymbol{\eta}) = \prod_{i=1}^{C} p(c_i \mid \eta_i) = \prod_{i=1}^{C} \text{Bernoulli}\big(c_i; \sigma(\eta_i)\big).$$

The concept distribution is then trained by minimizing the negative marginal log-likelihood of the observed concepts:

$$\mathcal{L}_c = -\log p(\boldsymbol{c} \mid \boldsymbol{x}) = -\log \int p(\boldsymbol{c} \mid \boldsymbol{\eta}) p_\phi(\boldsymbol{\eta} \mid \boldsymbol{x}) d\boldsymbol{\eta}.$$

Because this integral has no closed-form solution, SCBM approximates it using $M$ Monte Carlo samples $\boldsymbol{\eta}^{(m)} \sim \mathcal{N}(\boldsymbol{\mu}(\boldsymbol{x}), \boldsymbol{\Sigma}(\boldsymbol{x}))$. Together with the factorization of $p(\boldsymbol{c} \mid \boldsymbol{\eta})$, concept loss can be estimated as

$$\mathcal{L}_c = -\log \left( \frac{1}{M} \sum_{m=1}^{M} \exp \left[ -\sum_{i=1}^{C} \text{BCE}\big(c_i, \, \sigma(\eta_i^{(m)})\big) \right] \right),$$

where BCE is the Binary Cross-Entropy loss and $\sigma(\cdot)$ denotes the sigmoid function. Additional details on the Monte Carlo estimators for the concept loss are provided in Appendix A.

To mitigate information leakage, SCBMs follow the approach of Hard CBMs (Havasi et al., 2022) by passing sampled binary concept values rather than continuous logits to the target predictor $g_\psi$. During training, the straight-through Gumbel-Softmax trick (Jang et al., 2017; Maddison et al., 2017) is applied to provide a differentiable approximation for the discrete Bernoulli sampling. The target predictor is optimized by minimizing the negative log-likelihood of the target label $y$ given the sampled concepts:

$$\mathcal{L}_y \approx -\log \left( \frac{1}{M} \sum_{m=1}^{M} p_\psi(y \mid \boldsymbol{c}^{(m)}) \right) = \text{CE} \left( y, \frac{1}{M} \sum_{m=1}^{M} g_\psi(\boldsymbol{c}^{(m)}) \right),$$

where $\boldsymbol{c}^{(m)} \sim \text{Bernoulli}\big(\sigma(\boldsymbol{\eta}^{(m)})\big)$, $\boldsymbol{\eta}^{(m)} \sim \mathcal{N}(\boldsymbol{\mu}(\boldsymbol{x}), \boldsymbol{\Sigma}(\boldsymbol{x}))$, and CE denotes the Cross-Entropy loss.

Finally, SCBMs regularize the learned dependency structure by penalizing the off-diagonal entries of the precision matrix, following the graphical-lasso intuition (Friedman et al., 2008):

$$\mathcal{L}_{\text{reg}} = \sum_{i \neq j} \left| \left[ \boldsymbol{\Sigma}_\phi(\boldsymbol{x})^{-1} \right]_{ij} \right|.$$

The final training objective is

$$\mathcal{L} = \mathcal{L}_c + \lambda_y \mathcal{L}_y + \lambda_{\text{reg}} \mathcal{L}_{\text{reg}},$$

where $\lambda_y$ and $\lambda_{\text{reg}}$ control the relative weights of the target loss and the precision-matrix regularization.

### 3.2 Dependency-Aware Interventions

A key advantage of CBMs is that they support test-time interventions: when a user identifies an incorrectly predicted concept, the concept value can be replaced with a corrected value before recomputing the target prediction. In a standard CBM (Koh et al., 2020), intervening on one concept only changes that concept and leaves the remaining concept predictions fixed. This can make intervention inefficient when concepts are correlated. SCBMs address this limitation by updating the non-intervened concepts via the learned joint Gaussian distribution over concept logits. Let $\mathbb{S} \subset \{1, \ldots, C\}$ denote the set of intervened concepts and let $\mathbb{S}^{\complement}$ denote its complement. Given intervention logits $\boldsymbol{\eta}'_{\mathbb{S}}$ for the corrected concepts, the logits of the non-intervened concepts $\boldsymbol{\eta}_{\mathbb{S}^{\complement}}$ can be updated by conditioning the learned Gaussian distribution:

$$\boldsymbol{\eta}_{\mathbb{S}^{\complement}} \mid \boldsymbol{x}, \boldsymbol{\eta}'_{\mathbb{S}} \sim \mathcal{N}\left( \bar{\boldsymbol{\mu}}_{\mathbb{S}^{\complement}}, \overline{\boldsymbol{\Sigma}}_{\mathbb{S}^{\complement}\mathbb{S}^{\complement}} \right),$$

$$\bar{\boldsymbol{\mu}}_{\mathbb{S}^{\complement}} = \boldsymbol{\mu}_{\mathbb{S}^{\complement}} + \boldsymbol{\Sigma}_{\mathbb{S}^{\complement}, \mathbb{S}} \boldsymbol{\Sigma}_{\mathbb{S}, \mathbb{S}}^{-1} (\boldsymbol{\eta}'_{\mathbb{S}} - \boldsymbol{\mu}_{\mathbb{S}}),$$

$$\overline{\boldsymbol{\Sigma}}_{\mathbb{S}^{\complement}\mathbb{S}^{\complement}} = \boldsymbol{\Sigma}_{\mathbb{S}^{\complement},\mathbb{S}^{\complement}} - \boldsymbol{\Sigma}_{\mathbb{S}^{\complement},\mathbb{S}}\boldsymbol{\Sigma}_{\mathbb{S},\mathbb{S}}^{-1}\boldsymbol{\Sigma}_{\mathbb{S},\mathbb{S}^{\complement}}.$$

The remaining question is how to choose the intervention logits $\boldsymbol{\eta}'_{\mathbb{S}}$. In the standard CBM setting, Koh et al. (2020) set each intervention logit $\eta'_i$ to the empirical 95th percentile of the training logits when the corrected concept value is $c_i = 1$, and to the empirical 5th percentile when $c_i = 0$. Although this percentile-based strategy can be applied to SCBM, it can be suboptimal when interventions are propagated through the learned covariance structure (Vandenhirtz et al., 2024).

First, if the original predicted mean $\mu_i$ is already more extreme than the selected percentile, replacing $\eta_i$ with the fixed percentile can move the intervention logit in the wrong direction. This incorrect displacement is then propagated to the non-intervened logits through the conditional update of $\bar{\boldsymbol{\mu}}_{\mathbb{S}^{\complement}}$. As a result, the updated non-intervened logits may be shifted in an unintended direction. Second, choosing intervention logits too far from $\boldsymbol{\mu}_{\mathbb{S}}$ can make the conditional mean shift $\boldsymbol{\Sigma}_{\mathbb{S}^{\complement},\mathbb{S}}\boldsymbol{\Sigma}_{\mathbb{S},\mathbb{S}}^{-1}(\boldsymbol{\eta}'_{\mathbb{S}} - \boldsymbol{\mu}_{\mathbb{S}})$ dominate the original mean prediction for the non-intervened logits.

Vandenhirtz et al. (2024) therefore require the intervention logits to satisfy two conditions. First, each logit must move in the direction of the corrected concept value:

$$\eta_i - \mu_i \geq 0 \quad \text{if } c_i = 1, \qquad \eta_i - \mu_i \leq 0 \quad \text{if } c_i = 0.$$

Second, the intervention logits should remain plausible under the predicted Gaussian distribution. Since we model $\boldsymbol{\eta}$ as a multivariate Gaussian, the marginal distribution of $\boldsymbol{\eta}_{\mathbb{S}}$ for any subset $\mathbb{S}$ is also Gaussian:

$$\boldsymbol{\eta}_{\mathbb{S}} \sim \mathcal{N}(\boldsymbol{\mu}_{\mathbb{S}}, \boldsymbol{\Sigma}_{\mathbb{S},\mathbb{S}}).$$

By the property of multivariate Gaussian distributions, the squared Mahalanobis distance

$$d_M^2(\boldsymbol{\eta}_{\mathbb{S}}, \boldsymbol{\mu}_{\mathbb{S}}) = (\boldsymbol{\eta}_{\mathbb{S}} - \boldsymbol{\mu}_{\mathbb{S}})^{\top}\boldsymbol{\Sigma}_{\mathbb{S},\mathbb{S}}^{-1}(\boldsymbol{\eta}_{\mathbb{S}} - \boldsymbol{\mu}_{\mathbb{S}})$$

follows a Chi-squared distribution with $|\mathbb{S}|$ degrees of freedom. This gives the following Mahalanobis-distance constraint on the intervention logits:

$$d_M^2(\boldsymbol{\eta}_{\mathbb{S}}, \boldsymbol{\mu}_{\mathbb{S}}) \leq \chi_{|\mathbb{S}|,1-\alpha}^2,$$

where $\chi_{|\mathbb{S}|,1-\alpha}^2$ is the $(1-\alpha)$-quantile of the Chi-squared distribution with $|\mathbb{S}|$ degrees of freedom, and we use $\alpha = 0.01$ in our experiments. Combining these conditions, Vandenhirtz et al. (2024) formulate the selection of intervention logits as the following constrained optimization problem:

$$\boldsymbol{\eta}'_{\mathbb{S}} = \arg\max_{\boldsymbol{\eta}_{\mathbb{S}}} \log p(\boldsymbol{c}_{\mathbb{S}} \mid \boldsymbol{\eta}_{\mathbb{S}})$$

$$\text{s.t.} \quad \begin{cases} \eta_i \geq \mu_i, & \text{if } c_i = 1 \\ \eta_i \leq \mu_i, & \text{if } c_i = 0 \end{cases} \quad \forall i \in \mathbb{S},$$

$$\text{and} \quad (\boldsymbol{\eta}_{\mathbb{S}} - \boldsymbol{\mu}_{\mathbb{S}})^{\top}\boldsymbol{\Sigma}_{\mathbb{S},\mathbb{S}}^{-1}(\boldsymbol{\eta}_{\mathbb{S}} - \boldsymbol{\mu}_{\mathbb{S}}) \leq \chi_{|\mathbb{S}|,1-\alpha}^2.$$

When predicting the target label $\hat{y}'$ under interventions, the intervention logits $\boldsymbol{\eta}'_{\mathbb{S}}$ are obtained by solving the constrained optimization problem. The non-intervened logits $\boldsymbol{\eta}_{\mathbb{S}^{\complement}}$ are subsequently sampled from the resulting conditional Gaussian distribution. Finally, the non-intervened concepts sampled from these updated logits are combined with the ground-truth binary values of the intervened concepts, and this joint concept vector is used to predict the final target $\hat{y}'$.

### 3.3 Solver for Dependency-Aware Interventions

In the original work (Vandenhirtz et al., 2024), the constrained optimization problem is solved independently for each sample using the SLSQP algorithm (Kraft, 1988) through SciPy (Virtanen et al., 2020) on the CPU. Since intervention optimization must be performed repeatedly for all test samples, solving each constrained subproblem with a generic optimizer quickly becomes a bottleneck. To exploit GPU parallelism, we replace

**Batched Frank-Wolfe iteration**

```
# Select the intervened-concept subproblem.
# mu, direction, target: (B, C); A: (B, C, C)
mu, A, target, direction = get_intervened_marginals(batch_tensors)

# Initialize all samples on the ellipsoid boundary. u: (B, C)
u = init_on_ellipsoid_boundary(A, chi2)

for t in range(T):
    eta = mu + direction * u                        # (B, C)
    grad = direction * (sigmoid(eta) - target)      # (B, C)

    # Closed-form LMO on the ellipsoid. v: (B, C)
    v = solve(A, -grad)
    v = clamp(v, min=eps)
    v = rescale_to_boundary(v, A, chi2)

    # Batched grid line search on the BCE objective.
    weights = linspace(0, 1, K)                     # (K,), broadcasts as (1, K, 1)
    candidates = (1 - weights) * u + weights * v    # (B, K, C), u/v broadcast as (B, 1, C)
    losses = BCE(mu + direction * candidates, target)  # (B, K), target expand to (B, K, C)
    u = candidates[:, argmin(losses, dim=1)]        # (B, C)

eta = mu + direction * u
```

Figure 2: Pseudocode for the batched Frank-Wolfe solver used to compute SCBM intervention logits. The pseudocode variables `mu`, `A`, `target`, and `direction` correspond to $\boldsymbol{\mu}_{\mathbb{S}}$, $\boldsymbol{A}$, $\boldsymbol{c}_{\mathbb{S}}$, and $\boldsymbol{s}$, respectively; T denotes the number of Frank-Wolfe iterations, and K denotes the number of grid points used in the line search.

this per-sample CPU solver with a modified Frank–Wolfe solver implemented entirely in PyTorch (Paszke et al., 2019), enabling batched optimization across multiple intervention instances.

To simplify the direction constraints, we reparameterize the intervention logits as

$$\boldsymbol{\eta}'_{\mathbb{S}} = \boldsymbol{\mu}_{\mathbb{S}} + \boldsymbol{s} \odot \boldsymbol{u}, \qquad s_i = 2c_i - 1,$$

where $c_i$ denotes the known true value for intervened concept $i$, and $\odot$ denotes elementwise multiplication. With this reparameterization, the constraints $\eta_i \geq \mu_i$ when $c_i = 1$ and $\eta_i \leq \mu_i$ when $c_i = 0$ are both converted into the simple non-negativity constraint $\boldsymbol{u} \geq \boldsymbol{0}$. At the same time, the Mahalanobis-distance constraint becomes

$$\boldsymbol{u}^\top \boldsymbol{A} \boldsymbol{u} \leq \chi^2_{|\mathbb{S}|, 1-\alpha}, \qquad \boldsymbol{A} = \mathrm{diag}(\boldsymbol{s}) \boldsymbol{\Sigma}^{-1}_{\mathbb{S}, \mathbb{S}} \mathrm{diag}(\boldsymbol{s}).$$

The resulting feasible set is the intersection of an ellipsoid and the non-negative orthant. The original maximization of $\log p(\boldsymbol{c}_{\mathbb{S}} \mid \boldsymbol{\eta}'_{\mathbb{S}})$ is implemented as minimization of the binary cross-entropy objective

$$F(\boldsymbol{u}) = \mathrm{BCEWithLogits}(\boldsymbol{\mu}_{\mathbb{S}} + \boldsymbol{s} \odot \boldsymbol{u}, \boldsymbol{c}_{\mathbb{S}}),$$

whose gradient with respect to $\boldsymbol{u}$ is $\boldsymbol{s} \odot (\sigma(\boldsymbol{\mu}_{\mathbb{S}} + \boldsymbol{s} \odot \boldsymbol{u}) - \boldsymbol{c}_{\mathbb{S}})$, where $\sigma(\cdot)$ is the sigmoid function.

Since $F(\boldsymbol{u})$ is convex and the feasible set is compact and convex, we use the Frank-Wolfe algorithm (Frank & Wolfe, 1956), which only requires a linear minimization oracle (LMO) over the feasible set at each iteration. In our case, the exact LMO over the intersection of the ellipsoid and non-negativity constraints does not have a closed-form solution, so we approximate it with a clamp-and-rescale heuristic.

Figure 2 summarizes our solver. We initialize $\boldsymbol{u}$ by scaling the all-ones direction to the ellipsoid boundary. At iteration $t$, because the ellipsoid and non-negativity constraints do not jointly admit a simple closed-form LMO, the algorithm first solves the LMO for the ellipsoidal constraint alone. From the Lagrangian of this ellipsoid-only subproblem, the resulting oracle direction is proportional to $-\boldsymbol{A}^{-1} \nabla F(\boldsymbol{u}_t)$, where $\nabla F(\boldsymbol{u}_t)$ denotes the current gradient. The non-negativity constraint is then enforced approximately by clamping negative coordinates to a small positive value, after which the vector is rescaled back to the ellipsoid boundary, yielding a feasible point $\boldsymbol{v}_t$. A batched grid line search is subsequently performed between the current iterate $\boldsymbol{u}_t$ and $\boldsymbol{v}_t$: the objective $F$ is evaluated at $K$ points on the segment connecting $\boldsymbol{u}_t$ and $\boldsymbol{v}_t$, and the point achieving the lowest value is taken as the next iterate $\boldsymbol{u}_{t+1}$, which is guaranteed to be no worse than $\boldsymbol{u}_t$ since $\boldsymbol{u}_t$ itself is included among the candidates. Appendix D.5 provides a review of the classical Frank-Wolfe

algorithm, discusses how the clamp-and-rescale heuristic relates to the standard Frank-Wolfe guarantees, and presents empirical convergence validation against SLSQP.

## 4 Experimental Setup

### 4.1 Datasets and Evaluation

**Synthetic Dataset**  The synthetic dataset follows the correlated-concept construction of Vandenhirtz et al. (2024). It contains $N = 50,000$ samples with $C = 100$ binary concepts and $p = 1,500$ observed input features, split into train, validation, and test sets with a $60\%/20\%/20\%$ ratio. The dataset is designed as a controlled benchmark with correlated concepts, allowing us to test whether modeling concept dependencies improves predictive and intervention performance. The data generation procedure is summarized in Appendix B.1, which also describes the adjusted dataset hyperparameters used in our reproduction.

**Natural Image Datasets**  For the natural image experiments, we use the same datasets and concept-label sources as Vandenhirtz et al. (2024): the Caltech-UCSD Birds-200-2011 dataset (CUB-200-2011) (Wah et al., 2011) and CIFAR-10 (Krizhevsky et al., 2009). CUB-200-2011 contains 11,788 images from 200 bird species. The image classification task uses the 112 binary bird attributes selected in the original CBM work (Koh et al., 2020) as concepts and bird species as target labels. For CIFAR-10, following the setup of Vandenhirtz et al. (2024), we use a CLIP-based concept-labeling pipeline to reduce the need for manual concept annotations. The concept vocabulary consists of 143 concepts generated with GPT-3 in prior work on label-free CBMs (Brown et al., 2020; Oikarinen et al., 2023). Binary concept labels are generated before training by comparing CLIP (Radford et al., 2021) similarities between each image and paired positive and negative text prompts, such as "machine" and "not machine".

**Evaluation**  Following Vandenhirtz et al. (2024), we evaluate predictive performance and concept-probability calibration to compare models. Predictive performance is measured by concept accuracy and target accuracy on the test set. Calibration is measured using the Brier score (Glenn, 1950) and binary Expected Calibration Error (ECE) (Naeini et al., 2015; Kumar et al., 2019). For both calibration metrics, we treat each sample-concept pair as a Bernoulli probability prediction and average over all sample-concept pairs. The full definitions of the Brier score and ECE used in our experiments are provided in Appendix B.2.

### 4.2 Baselines

Following the comparison setup of Vandenhirtz et al. (2024), we compare SCBM with Hard CBM, autoregressive CBM (Havasi et al., 2022), and CEM (Espinosa Zarlenga et al., 2022). These baselines cover three relevant alternatives to SCBM: a hard binary bottleneck without dependency-aware updates, an order-dependent approach to modeling concept correlations, and an embedding-based concept bottleneck. The model comprises a backbone for concept prediction and a linear target head that maps concept representations to target predictions. Detailed hyperparameters and model architectures are provided in Appendix C.

For SCBM, we evaluate two covariance parameterizations considered by Vandenhirtz et al. (2024): an amortized variant, where the covariance matrix $\Sigma(\boldsymbol{x})$ is predicted for each input, and a global variant, where a single covariance matrix $\Sigma$ is learned for the entire dataset. The global variant may be preferable in settings where a fixed concept dependency structure is a strong prior.

## 5 Main Results

### 5.1 Test Performance

In Table 1, we report predictive performance and calibration metrics on the test set before intervention. In terms of concept and target accuracy, SCBM performs competitively with the baselines, and no model consistently dominates across all datasets. This suggests that modeling concept dependencies does not come at a clear cost in predictive performance.

Table 1: Test-set performance across ten random seeds before intervention, reported as mean ± sample standard deviation. Concept accuracy, target accuracy, Brier score, and ECE are reported as percentages. **Bold** and underlined mean values indicate the best and second-best results for each metric and dataset, respectively; higher is better for accuracies, and lower is better for Brier score and ECE.

| Dataset | Model | Concept Accuracy | Target Accuracy | Brier | ECE |
|---|---|---|---|---|---|
| Synthetic | Hard CBM | $69.33 \pm 0.07$ | $66.76 \pm 0.24$ | $20.02 \pm 0.04$ | $5.89 \pm 0.14$ |
| | CEM | $69.37 \pm 0.08$ | $66.79 \pm 0.37$ | $20.08 \pm 0.04$ | $6.16 \pm 0.09$ |
| | Autoregressive CBM | $\mathbf{70.66} \pm 0.03$ | $67.28 \pm 0.15$ | $\mathbf{18.97} \pm 0.02$ | $2.34 \pm 0.09$ |
| | Global SCBM | $70.50 \pm 0.04$ | $\underline{67.43} \pm 0.20$ | $19.04 \pm 0.02$ | $\underline{2.31} \pm 0.07$ |
| | Amortized SCBM | $\underline{70.53} \pm 0.05$ | $\mathbf{67.47} \pm 0.13$ | $\underline{19.01} \pm 0.02$ | $\mathbf{2.11} \pm 0.07$ |
| CUB | Hard CBM | $94.83 \pm 0.07$ | $67.11 \pm 0.68$ | $4.02 \pm 0.08$ | $2.48 \pm 0.13$ |
| | CEM | $94.94 \pm 0.09$ | $68.79 \pm 0.66$ | $4.17 \pm 0.08$ | $3.34 \pm 0.10$ |
| | Autoregressive CBM | $\mathbf{95.24} \pm 0.09$ | $\underline{68.83} \pm 0.58$ | $\underline{3.84} \pm 0.07$ | $2.82 \pm 0.10$ |
| | Global SCBM | $94.79 \pm 0.11$ | $67.10 \pm 0.76$ | $4.01 \pm 0.09$ | $\underline{2.41} \pm 0.09$ |
| | Amortized SCBM | $\underline{95.13} \pm 0.10$ | $\mathbf{69.44} \pm 0.59$ | $\mathbf{3.70} \pm 0.08$ | $\mathbf{1.90} \pm 0.09$ |
| CIFAR-10 | Hard CBM | $85.48 \pm 0.07$ | $69.72 \pm 0.20$ | $10.50 \pm 0.06$ | $5.29 \pm 0.12$ |
| | CEM | $85.23 \pm 0.17$ | $\mathbf{72.17} \pm 0.61$ | $10.84 \pm 0.22$ | $6.46 \pm 0.55$ |
| | Autoregressive CBM | $85.57 \pm 0.10$ | $69.56 \pm 0.53$ | $10.40 \pm 0.12$ | $5.23 \pm 0.35$ |
| | Global SCBM | $\underline{85.87} \pm 0.03$ | $70.89 \pm 0.37$ | $\underline{9.98} \pm 0.02$ | $\underline{3.15} \pm 0.07$ |
| | Amortized SCBM | $\mathbf{86.08} \pm 0.07$ | $\underline{71.97} \pm 0.26$ | $\mathbf{9.78} \pm 0.05$ | $\mathbf{1.96} \pm 0.22$ |

We also observe that the amortized SCBM variant consistently outperforms the global covariance variant across the three datasets, suggesting that the additional flexibility of instance-wise covariance prediction may be useful in these settings. In contrast, the global covariance variant provides a single dataset-level dependency structure that is easier to inspect. For example, the selected body-pattern attributes in Figure 3a show interpretable correlation patterns: striped-pattern attributes are positively correlated with each other, solid-pattern attributes are positively correlated with each other, and striped patterns are negatively correlated with solid patterns.

On CIFAR-10, we observe a dataset-specific discrepancy. Although CEM has the lowest concept accuracy among the compared models, it achieves the highest target accuracy. This observation is consistent with the information-leakage discussion in Section 2: high-dimensional concept embeddings may carry task-relevant information beyond the binary concept probabilities. This suggests an expressiveness-faithfulness trade-off in concept-based models: richer concept representations can improve target performance, but they may weaken the extent to which the target prediction can be understood solely through concept probabilities.

One possible reason that this discrepancy appears on CIFAR-10 but not on the other datasets is the sufficiency of the concept bottleneck. CUB uses human-annotated visual attributes, and the synthetic dataset is constructed so that the target is determined by the concepts. In contrast, the CIFAR-10 concepts are generated by GPT-3 (Brown et al., 2020; Oikarinen et al., 2023), and the binary concept set may provide a weaker description of the target classes. In this setting, models whose target predictors rely on binary concept predictions may be more constrained by the incompleteness of the concept set. CEM's richer embeddings, in contrast, may provide additional task-relevant information to the target predictor, yielding higher target accuracy despite lower scalar concept accuracy.

The calibration results further support the benefit of SCBM's explicit distributional parameterization of concepts. In Table 1, amortized SCBM achieves the best ECE across all three datasets and the best or second-best Brier score, suggesting more reliable concept-probability estimates. Reliable concept uncertainty estimates are particularly relevant for intervention. At each intervention step, the uncertainty-based policy selects the concept with the highest predictive uncertainty, aiming to improve intervention efficiency by correcting uncertain concepts earlier. Here, uncertainty is measured by the proximity of the predicted concept probability to 0.5; that is, the concept with the smallest $|\hat{p} - 0.5|$ is selected for intervention. We

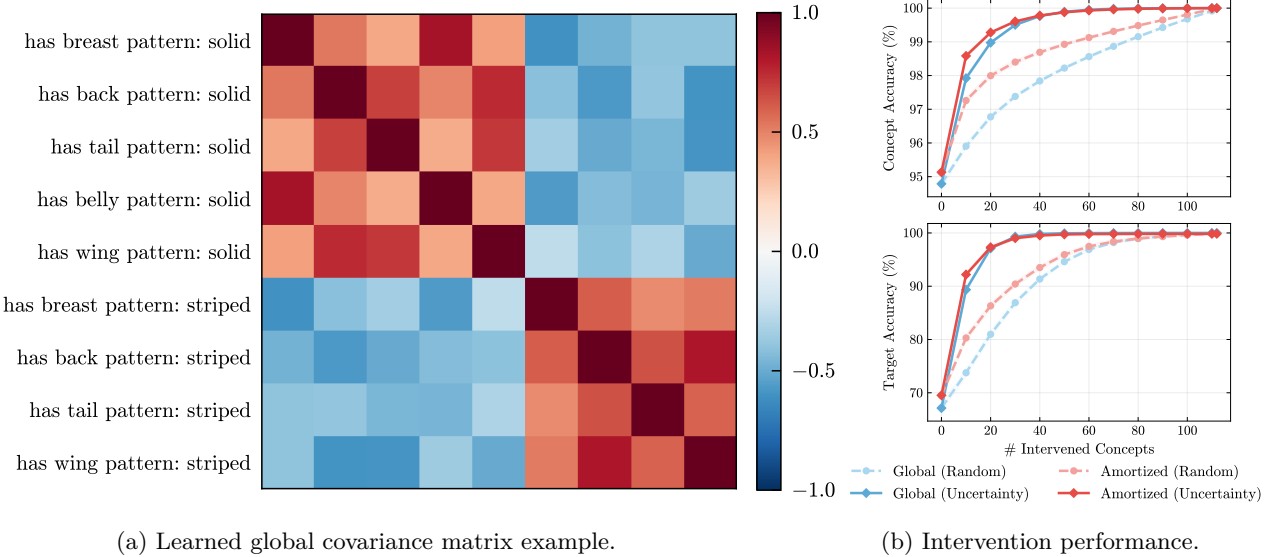

(a) Learned global covariance matrix example.    (b) Intervention performance.

Figure 3: CUB analysis of learned concept dependencies and uncertainty-guided interventions. **(a)** A subset of the global covariance matrix learned by Global SCBM, illustrating dataset-level dependency structure among selected concepts. **(b)** Intervention performance of SCBMs measured in concept and target accuracy (%) on CUB for random and uncertainty-based policy.

compare random and uncertainty-based intervention policies for SCBM on CUB. As shown in Figure 3b, the substantial gap between the two policies indicates that SCBM's predicted concept probabilities carry meaningful uncertainty information for guiding interventions.

## 5.2 Intervention

Figure 4 shows intervention performance across datasets under the uncertainty-based intervention policy described above. The curves report concept and target accuracy after intervening on an increasing number of concepts. For concept prediction, SCBM variants generally show steeper improvement than the baselines, indicating that each manual correction improves not only the intervened concept but also related non-intervened concepts. This behavior is consistent with the dependency-aware intervention mechanism introduced in Section 3.2, which propagates intervention information to correlated concepts through the learned dependency structure. Similarly, the target accuracy curves show that the improved concept predictions also translate to improvements in target prediction. Notably, in the most practical scenario with a small number of interventions, SCBM consistently outperforms the baselines in both concept and target accuracy, indicating the value of modeling concept dependencies for efficient and effective interventions.

The two SCBM variants behave differently across datasets. On the natural image datasets, amortized SCBM generally outperforms global SCBM, consistent with the test-set results in Table 1. This suggests that input-dependent covariance prediction can be useful when concept dependencies vary across instances. On the synthetic dataset, global SCBM performs slightly better, which is consistent with the data generation process having a fixed covariance structure shared across samples.

Autoregressive CBM also models concept dependencies and therefore improves over Hard CBM in intervention as expected. However, its advantage over SCBM appears mainly at larger intervention budgets in the target accuracy curves. This setting is less aligned with the intended use case of concept interventions, where the goal is to obtain large performance gains from only a small number of manual corrections. The better performance of autoregressive CBM at larger intervention budgets may partly benefits from the independent training procedure used for autoregressive CBMs, where the target predictor is trained using ground-truth concept labels rather than predicted concepts. Finally, CEM shows weaker target intervention performance in our experiments, which may be explained by the omission of the intervention-specific optimization from

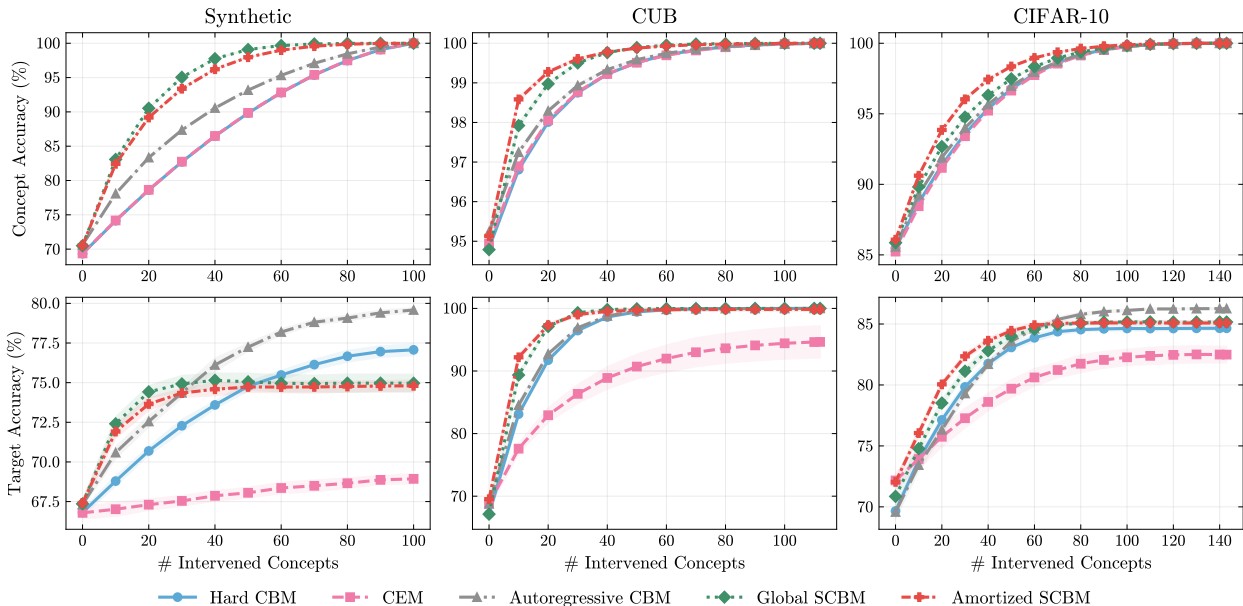

Figure 4: Intervention performance across datasets and models. The x-axis shows the number of intervened concepts, and the y-axis shows accuracy after intervention. Curves are averaged over ten random seeds, and shaded areas indicate $\pm$ one sample standard deviation.

the original CEM framework (Espinosa Zarlenga et al., 2022), as Vandenhirtz et al. (2024) exclude it to maintain a fair comparison across models.

# 6 Practical Reproduction Findings

In this section, we report the practical cost of reproducing SCBM, the optimizations we made to reduce this cost, and their implications for the runtime comparisons reported in the original paper.

## 6.1 Optimizing the Reproduction Pipeline

Vandenhirtz et al. (2024) report that reproducing the full set of experiments with the original implementation requires approximately 1,200 GPU hours on GeForce RTX 2080 hardware. To make the reproduction feasible within our compute budget, we ran our experiments on a GeForce RTX 4090. However, our initial profiling showed that the original implementation remained slow even on stronger hardware, with low and unstable GPU utilization. Through iterative profiling and optimization, we identified overhead from several parts of the pipeline, including CPU-bound data loading, unnecessary metric computation during training, model-forward overhead from naive for loops, and repeated intervention-time CPU–GPU data transfer.

We addressed these bottlenecks through four types of optimizations. First, we optimized the data pipeline by moving suitable transformations to batched GPU execution. Second, we vectorized model execution while preserving the original computation, including batched concept-embedding computation for CEM, parallel teacher-forced concept prediction for autoregressive CBM, and batched Monte Carlo samples prediction for all models requiring Monte Carlo sampling. Third, we changed intervention evaluation from a step-first order to a batch-first order, so that intermediate intervention states remain on the GPU across intervention steps. Fourth, for SCBM dependency-aware interventions, we replaced the per-sample CPU-based SLSQP solver (Kraft, 1988; Virtanen et al., 2020) with a batched Frank-Wolfe solver (Frank & Wolfe, 1956) implemented in PyTorch, which produces numerically close solutions while substantially reducing solver runtime. Further implementation details and validation results are provided in Appendix D.

Table 2: Timing comparison between the legacy and optimized implementations on CIFAR-10.

| Dataset | Model | Stage | Legacy (s) | Optimized (s) | Speedup |
|---|---|---|---|---|---|
| CIFAR-10 | Autoregressive CBM | pretrain | 188.2 | 8.7 | 21.6× |
| | | concept | 198.4 | 11.4 | 17.4× |
| | | target | 111.8 | 3.6 | 31.1× |
| | | intervention | 1264.6 | 86.1 | 14.7× |
| | CEM | joint | 263.3 | 12.6 | 20.9× |
| | | intervention | 185.5 | 74.5 | 2.5× |
| | Hard CBM | joint | 112.8 | 14.0 | 8.1× |
| | | intervention | 193.3 | 73.3 | 2.6× |
| | Amortized SCBM | joint | 120.2 | 29.3 | 4.1× |
| | | intervention | 278.4 | 104.8 | 2.7× |
| | Global SCBM | joint | 124.6 | 27.5 | 4.5× |
| | | intervention | 504.8 | 104.2 | 4.8× |

These optimizations substantially reduce the practical cost of reproduction. On a single GeForce RTX 4090, we completed the reproduction in approximately 62 wall-clock hours. In Table 2, we compare the runtime on the CIFAR-10 dataset between the legacy and optimized implementations using a single GeForce RTX 4090, ensuring that the reported speedups strictly isolate the effect of our optimizations. The benchmark is reported by model and stage. For the autoregressive CBM, *pretrain* denotes pretraining of the autoregressive concept predictors, *concept* denotes training of the encoder and concept predictors, and *target* denotes training of the target head. For the other models, *joint* denotes joint training of the encoder, concept predictor, and target head. The *intervention* stage denotes intervention evaluation. Training-stage runtimes are measured over five training epochs using the hyperparameters in Table 3. Intervention runtimes include initialization and intervention evaluation over two batches of 64 samples.

## 6.2 Revisiting the Runtime Comparison

These optimizations also change the relative runtime behavior of the compared models. Vandenhirtz et al. (2024) report that SCBM is much more efficient than autoregressive CBM during evaluation, and has comparable training time with other baselines. However, the optimized benchmark in Table 2 shows a different runtime ordering during intervention evaluation. With the optimizations described above, autoregressive CBM can achieve intervention-evaluation runtimes comparable to SCBM, and SCBM's training time is often longer than other baselines due to the additional computational overhead of learning and sampling from the joint Gaussian distribution. This pattern also holds on the other datasets we evaluated (Appendix D.6), suggesting that runtime comparisons should be interpreted as properties of a method–implementation pair rather than of the method alone. Therefore, our optimized implementation does not support the original conclusion that SCBM is more efficient than autoregressive CBM during evaluation. Instead, our findings suggest that SCBM's practical advantage is better supported by its ability to exploit dependencies among concepts during intervention than by raw computational efficiency.

## 7 Conclusion

In this reproduction study, we reproduced the key experiments of the SCBM framework proposed by Vandenhirtz et al. (2024) on a synthetic dataset and two natural image datasets. Overall, our results are broadly consistent with the original paper and support the reproducibility of its main empirical claims. In particular, SCBM models concept dependencies without a clear degradation in predictive performance before intervention, and achieves stronger concept-probability calibration than the compared baselines. The global SCBM variant also provides a valuable tool for inspecting dataset-level concept relationships.

Our intervention experiments further support the main motivation of SCBM: modeling concept dependencies can make concept interventions more efficient. By propagating the effect of a corrected concept to correlated non-intervened concepts, SCBM improves concept and target accuracy with fewer manual interventions. We also find that SCBM benefits from uncertainty-based intervention policies, which prioritize concepts whose predicted probabilities are closest to 0.5, suggesting that SCBM's concept-probability estimates carry useful uncertainty signals for selecting interventions and reducing the need for users to manually search through the full concept set.

Beyond empirical reproduction, we examined the practical cost of reproducing SCBM. Starting from the official implementation, we identified several implementation bottlenecks and refactored the experimental pipeline to reduce avoidable overhead while preserving the intended model definitions and evaluation protocol. This substantially reduced the practical reproduction cost on modern hardware and made it possible to complete the experiments in approximately 62 wall-clock hours on a single GeForce RTX 4090, making it more feasible for future work to explore additional baselines, ablations, or robustness checks within a similar compute budget.

We also revisited the computational-efficiency interpretation of SCBM. After optimization, the relative runtime behavior of SCBM and the baselines differs from that reported with the original implementation, indicating that runtime comparisons are sensitive to implementation choices. The more robust practical advantage of SCBM is its ability to exploit dependencies among concepts during intervention without imposing a fixed concept ordering. This contrasts with autoregressive CBMs, whose ordering may not correspond to a natural semantic or causal structure in the data.

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

# A Monte Carlo Estimators for Concept Loss

To learn the concept distribution, SCBM minimizes the negative marginal log-likelihood of the observed concept labels. For a single data point $(\boldsymbol{x}, \boldsymbol{c})$, the concept loss is

$$\mathcal{L}_c = -\log p(\boldsymbol{c} \mid \boldsymbol{x}) = -\log \int p(\boldsymbol{c} \mid \boldsymbol{\eta}) p_\phi(\boldsymbol{\eta} \mid \boldsymbol{x}) d\boldsymbol{\eta},$$

where $p_\phi(\boldsymbol{\eta} \mid \boldsymbol{x})$ denotes the multivariate Gaussian $\mathcal{N}(\boldsymbol{\mu}(\boldsymbol{x}), \boldsymbol{\Sigma}(\boldsymbol{x}))$ parameterized by the neural network encoder with parameters $\phi$. This integral is intractable due to the sigmoid-transform applied to the concept logits in the Bernoulli likelihood. SCBM therefore approximates the marginal likelihood using $M$ Monte Carlo samples: $\boldsymbol{\eta}^{(m)} \sim \mathcal{N}(\boldsymbol{\mu}(\boldsymbol{x}), \boldsymbol{\Sigma}(\boldsymbol{x}))$:

$$\mathcal{L}_c \approx -\log \left( \frac{1}{M} \sum_{m=1}^{M} p(\boldsymbol{c} \mid \boldsymbol{\eta}^{(m)}) \right).$$

To enable end-to-end backpropagation through this sampling process, SCBMs employ the reparameterization trick:

$$\boldsymbol{\eta}^{(m)} = \boldsymbol{\mu}(\boldsymbol{x}) + \boldsymbol{L}(\boldsymbol{x})\boldsymbol{\epsilon}^{(m)}, \qquad \boldsymbol{\epsilon}^{(m)} \sim \mathcal{N}(\boldsymbol{0}, \boldsymbol{I}),$$

where $\boldsymbol{L}(\boldsymbol{x})$ is the Cholesky factor of $\boldsymbol{\Sigma}(\boldsymbol{x})$, so that $\boldsymbol{L}(\boldsymbol{x})\boldsymbol{L}(\boldsymbol{x})^T = \boldsymbol{\Sigma}(\boldsymbol{x})$.

Using the conditional independence assumption, the concept loss can be rewritten in log-space directly in terms of the Binary Cross-Entropy (BCE) loss for each concept:

$$\begin{aligned}
\mathcal{L}_c &\approx -\log \left( \frac{1}{M} \sum_{m=1}^{M} p(\boldsymbol{c} \mid \boldsymbol{\eta}^{(m)}) \right) \\
&= -\log \left( \frac{1}{M} \sum_{m=1}^{M} \exp \sum_{i=1}^{C} \log p(c_i \mid \eta_i^{(m)}) \right) \\
&= -\log \left( \frac{1}{M} \sum_{m=1}^{M} \exp \left[ -\sum_{i=1}^{C} \mathrm{BCE}(c_i, \sigma(\eta_i^{(m)})) \right] \right).
\end{aligned}$$

This factorization allows all concepts to be sampled in parallel for a given logit vector, enabling efficient joint training of both the concept and target predictors. In contrast, in the autoregressive CBM (Havasi et al., 2022), concept sampling is sequential and the target predictor is trained independently for efficiency.

# B Dataset and Evaluation Details

## B.1 Synthetic Dataset Generation

The synthetic data generation procedure used by Vandenhirtz et al. (2024) is summarized as follows. Let $\{(\boldsymbol{x}_n, \boldsymbol{c}_n, y_n)\}_{n=1}^{N}$ denote the dataset, where $\boldsymbol{x}_n \in \mathbb{R}^p$ is the observed input feature vector, $\boldsymbol{c}_n \in \{0,1\}^C$ is the binary concept vector, and $y_n$ is the target label. We use $N = 50{,}000$, $p = 1{,}500$, and $C = 100$, with a $60\%/20\%/20\%$ train/validation/test split. Concept dependencies are generated through correlated latent logits. The concept-logit covariance matrix is sampled in low-rank-plus-diagonal form as

$$\boldsymbol{\Sigma} = \boldsymbol{W}\boldsymbol{W}^T + \boldsymbol{D},$$

where $\boldsymbol{W} \in \mathbb{R}^{C \times 10}$ has entries sampled from $\mathcal{N}(0,1)$ and $\boldsymbol{D} = \mathrm{diag}(\delta_1, \ldots, \delta_C)$ with $\delta_i \sim \mathcal{U}[0,1]$. For each sample, latent logits are drawn as

$$\boldsymbol{\eta}_n \sim \mathcal{N}(\boldsymbol{0}, \boldsymbol{\Sigma}),$$

and binary concepts are obtained by thresholding the logits:

$$c_{n,i} = \mathbf{1}\{\eta_{n,i} \geq 0\}.$$

Observed input features are generated by applying a randomly initialized MLP $f : \mathbb{R}^C \to \mathbb{R}^p$ with two hidden layers of width $d_h$ and ReLU activations to the latent logits, followed by additive Gaussian noise:

$$\tilde{\boldsymbol{x}}_n = f(\boldsymbol{\eta}_n) + \boldsymbol{\epsilon}_n, \qquad \boldsymbol{\epsilon}_n \sim \mathcal{N}(\boldsymbol{0}, \delta \boldsymbol{I}_p).$$

Following Vandenhirtz et al. (2024), the MLP outputs are normalized across the full dataset before adding Gaussian noise, and the noisy features are normalized again across the full dataset to obtain the final observed features $\boldsymbol{x}_n$. The original codebase uses $d_h = 5$ and $\delta = 1$. However, we found that this setting makes the dataset too noisy for all models to learn meaningful signals, so we set $d_h = 10$ to preserve more information in $\boldsymbol{x}$ and $\delta = 0.5$ to reduce the noise level.

Finally, target labels are generated by applying a randomly initialized linear map $g : \mathbb{R}^C \to \mathbb{R}$ to the binary concept vectors. Let $s_n = g(\boldsymbol{c}_n)$ and let $s_{\text{med}}$ denote the median of $\{s_n\}_{n=1}^N$. The target label is defined as

$$y_n = \mathbf{1}\{s_n \geq s_{\text{med}}\},$$

which yields a balanced binary classification task.

### B.2  Calibration Metrics

Let $N$ denote the number of evaluation samples and $C$ the number of concepts. Let $c_{n,j} \in \{0, 1\}$ denote the ground-truth value of concept $j$ for sample $n$, and let $\hat{p}_{n,j}$ denote the predicted probability that this concept is present. We evaluate calibration by treating each sample-concept pair $(n, j)$ as a Bernoulli probability prediction.

The Brier score measures the mean squared error between predicted concept probabilities and binary concept labels:

$$\text{Brier} = \frac{1}{NC} \sum_{n=1}^N \sum_{j=1}^C (\hat{p}_{n,j} - c_{n,j})^2.$$

ECE measures the same probabilities from a binned calibration perspective. We divide $[0, 1]$ into $K = 10$ uniformly spaced intervals $\{\mathcal{I}_k\}_{k=1}^K$ and define

$$\mathcal{B}_k = \{(n, j) : \hat{p}_{n,j} \in \mathcal{I}_k\}.$$

For each non-empty bin, we compute the empirical frequency of positive concept labels, denoted by $\text{freq}_k$, and the average predicted probability of concept presence, denoted by $\text{conf}_k$:

$$\text{freq}_k = \frac{1}{|\mathcal{B}_k|} \sum_{(n,j) \in \mathcal{B}_k} c_{n,j}, \qquad \text{conf}_k = \frac{1}{|\mathcal{B}_k|} \sum_{(n,j) \in \mathcal{B}_k} \hat{p}_{n,j}.$$

The ECE is then computed as:

$$\text{ECE} = \sum_{k=1}^K \frac{|\mathcal{B}_k|}{NC} \left| \text{freq}_k - \text{conf}_k \right|.$$

## C  Model Configurations and Hyperparameters

We summarize the model configurations and training hyperparameters used in our experiments. Each experiment is run over ten random seeds. All models are trained with the Adam optimizer (Kingma & Ba, 2014) and weight decay $10^{-4}$. To ensure a fair comparison between different models, all models use the same dataset-specific encoder architecture and a linear target head that maps the model's concept representation to target predictions. The dataset-specific training settings and encoder choices are summarized in Table 3.

The synthetic encoder is a three-block fully connected network, where each block consists of a linear layer, batch normalization, ReLU activation, and dropout. The CIFAR-10 encoder is a small CNN with two

Table 3: Dataset-specific training settings and encoder architectures.

| Dataset | Epochs | Batch size | Learning rate | Encoder |
|---------|--------|------------|---------------|---------|
| Synthetic | 150 | 256 | $10^{-4}$ | Three-block fully connected network |
| CUB | 300 | 64 | $10^{-4}$ | ResNet18 (He et al., 2016) |
| CIFAR-10 | 300 | 256 | $2 \times 10^{-4}$ | Small CNN |

convolutional layers, ReLU activations, and max pooling, followed by dropout and a linear layer. The CUB experiments use a ResNet18 encoder (He et al., 2016).

The autoregressive CBM uses an additional 50-epoch pretraining stage for the concept predictors to obtain a stable initialization. Sampling-based methods use $M = 100$ Monte Carlo samples. Amortized SCBM uses $\lambda_y = 1.0$ and $\lambda_{\text{reg}} = 1.0$, while Global SCBM uses $\lambda_y = 1.0$ and $\lambda_{\text{reg}} = 0.0$.

## D    Refactoring and Optimization Details

In this section, we summarize the main optimization categories and report benchmarks comparing the original and optimized implementations on our hardware, isolating the contribution of the implementation changes from hardware differences.

### D.1    General Refactoring

The original implementation tightly couples model-specific training logic, stage control, metric computation, and model-forward logic. As a first step, we refactored the training pipeline into a more modular structure. We implemented a staged experiment runner that makes model-specific training stages explicit, including the pretraining, concept-predictor training and target-head training of autoregressive CBM, and joint training for the other models. We also introduced model adapters to expose a consistent interface across models, allowing the same runner to apply consistent freezing policies, optimizer construction, and metric computation.

We also reorganized the model-forward logic. The legacy implementation keeps multiple baseline variants inside a single model wrapper and dispatches behavior through many if-else statements depending on the configuration. To preserve compatibility with the original interface, we did not fully split the implementation into separate model classes. Instead, we factored model-specific logic into separate methods while retaining the outer wrapper for input dispatch and output collection. This compromise improved readability and reduced duplicated conditional logic while keeping the refactoring scope manageable.

Finally, we simplified the metrics computed during training. We retained lightweight concept and target accuracy for training-time monitoring, while reserving more expensive calibration metrics, such as Brier score and ECE, for validation and test evaluation. This reduced avoidable CPU overhead during training without changing the evaluation protocol.

### D.2    Data Pipeline Optimizations

Profiling showed that the natural-image experiments were often limited by CPU-side data processing. In particular, repeated disk reads, PIL image decoding, and image transformations such as resizing, cropping, and data augmentation caused unstable GPU utilization. We therefore optimized the data pipeline to reduce repeated CPU work while preserving the preprocessing semantics of the original implementation.

For CIFAR-10, the full dataset is small enough to fit in memory, eliminating disk reads as a bottleneck. The remaining CPU overhead mainly comes from repeated CPU-side transform applied on every iteration. We address this by adopting the `torchvision` (maintainers & contributors, 2016) v2 transform API, which allows batch-level transforms to be executed on the GPU, leaving only minimal logic on the CPU side.

For CUB, a caching strategy implemented by the community[1] eliminates the dominant overhead from repeated disk reads and PIL decoding. For the remaining transforms, because CUB images vary in original resolution, each sample must first be individually loaded and cropped to a uniform size via `RandomResizedCrop` before batching is possible. This per-sample operation depends on the original image geometry and must remain on the CPU. Other transforms such as augmentation and normalization are moved to the GPU. As a result, CUB remains more CPU-bound than the synthetic and CIFAR-10 datasets, but the optimized pipeline still reduces avoidable data-loading overhead.

### D.3 Model-Level Optimizations

During initial experimental runs, we observed that CEM and the autoregressive CBM were substantially slower than Hard CBM. Inspecting the legacy implementation showed that both models evaluated many concept-specific modules sequentially in Python loops, creating avoidable model-forward overhead. Figure 5 illustrates these two model-level optimizations in simplified Python-style pseudocode.

For CEM, the positive and negative concept embedding layers were implemented as many small per-concept modules, as shown in the top panel of Figure 5. We replaced these repeated per-concept calls with two batched linear projections while preserving the original positive/negative embedding scoring semantics. This reduces Python-loop overhead and allows the concept embeddings for all concepts to be computed in a single batched tensor operation.

For the autoregressive CBM, the legacy implementation creates $C$ separate concept predictors and evaluates them in a Python loop. During teacher-forced concept training, the input to the predictor for concept $c_i$ is the intermediate representation together with the ground-truth previous concepts $c_1, \ldots, c_{i-1}$. Since these previous concepts are known during training, the concept predictors do not need to be evaluated sequentially. We therefore implemented a packed autoregressive concept predictor, illustrated in the bottom panel of Figure 5, that treats the concept dimension as a batch dimension during teacher-forced concept training. Each concept-specific MLP is represented by a slice of a batched parameter tensor, and all predictors use a padded input of size $d + C$, where $d$ is the dimension of the intermediate representation. A lower-triangular concept mask ensures that the predictor for concept $c_i$ only receives the previous concepts $c_{<i}$, preserving the autoregressive conditioning structure. This allows all concept predictors to be evaluated in parallel using batched matrix multiplications, reducing Python-loop and repeated concatenation overhead during concept training. For stages that require sequential sampling from the autoregressive model, we retain a single-concept forward path, since later sampled concepts depend on earlier sampled concepts.

For SCBM, we reduced overhead in the covariance-related computations. We cached the triangular indices used to construct covariance factors, avoiding repeated index construction during training and intervention. We also replaced explicit matrix inversions in the precision-matrix regularization loss with triangular solves based on the Cholesky factor where applicable.

Finally, for sampling-based methods, we vectorized target prediction across Monte Carlo samples. Instead of iterating over samples in a Python loop, we reshape the sampled concept tensors and evaluate the target predictor in batched form. This optimization is used during both training and intervention evaluation.

### D.4 Intervention Evaluation Optimizations

In the legacy implementation, intervention results are computed in a step-first order, as shown in the left panel of Figure 6. First, the initial intervention tensors are collected through a full model pass over the test set. For each intervention step, the legacy implementation iterates over the full test set, materializes the updated intermediate tensors, and uses them to construct the input dataset for the next intervention step. For $K$ intervention steps, this step-first order therefore performs $K$ dataset reconstructions, $K$ CPU-to-GPU transfers of the rolling intermediate state, and $K$ GPU-to-CPU materializations of the updated state. We therefore implemented intervention in a batch-first order. After the same initial collection step, the optimized implementation, shown in the right panel of Figure 6, processes one batch at a time and completes all intervention steps for that batch so that the relevant tensors remain on the GPU, therefore

---

[1]https://medium.com/@maxwbuckley/making-stochastic-concept-bottleneck-models-go-brrrr-with-birds-74dacba4975f

**CEM concept-embedding projections**

| Legacy | Optimized |
|---|---|

```
# h: [B, d]

# C small embedding projections for each sign.
pos_layers = ModuleList([Linear(d, E) for _ in range(C)])
neg_layers = ModuleList([Linear(d, E) for _ in range(C)])

# Per-concept embedding projections.
c_pos = [pos_layers[i](h) for i in range(C)]
c_neg = [neg_layers[i](h) for i in range(C)]

# Shared scorer, called once per concept.
p = [score(concat(c_pos[i], c_neg[i])) for i in range(C)]

# Final concept embeddings are also built per concept.
z = [
    p[i] * c_pos[i] + (1 - p[i]) * c_neg[i] for i in range(
      C)
]
z = concat(z, dim=1)  # [B, C * E]
```

```
# h: [B, d]

# Change: per-concept embedding layers -> two wide layers.
pos_projection = Linear(d, C * E)
neg_projection = Linear(d, C * E)

# Change: reshape the wide output to concept axis.
c_pos = pos_projection(h).reshape(B, C, E)  # [B, C, E]
c_neg = neg_projection(h).reshape(B, C, E)  # [B, C, E]

# Same shared scorer, applied over [B, C, 2E].
pairs = concat(c_pos, c_neg, dim=-1)
p = score(pairs)  # [B, C, 1]

z = p * c_pos + (1 - p) * c_neg
z = z.reshape(B, C * E)  # [B, C * E]
```

**Autoregressive CBM teacher-forced concept training**

| Legacy | Optimized |
|---|---|

```
# h: [B, d], c_true: [B, C]

# C predictors with growing input sizes.
predictors = ModuleList([
    Sequential(Linear(d + i, H),
               Linear(H, 1))
    for i in range(C)
])

# C separate forward calls, one per concept.
logits = []
for i in range(C):
    x_i = concat(h, c_true[:, :i])
    logits.append(predictors[i](x_i))

logits = concat(logits, dim=1)  # [B, C]
```

```
# h: [B, d], c_true: [B, C]

# Change: C sequential forward calls ->
# 2 batched matmuls over all C at once.
W1 = Parameter(shape=[C, d + C, H])
W2 = Parameter(shape=[C, H, 1])

# mask[i, j] = 1 only when j < i.
mask = lower_triangular(shape=[C, C])

# Change: build all concept inputs in parallel.
features = h[:, None, :].expand(-1, C, -1)
previous = c_true[:, None, :].expand(-1, C, -1)

# Keep only previous concepts for each predictor.
previous = previous * mask
x = concat(features, previous, dim=2)  # [B, C, d + C]

# Change: treat concept index C as batch dimension.
x = transpose(x, from=[B, C, d + C], to=[C, B, d + C])
hidden = bmm(x, W1)        # [C, B, H]
logits = bmm(hidden, W2)   # [C, B, 1]
logits = transpose(logits, to=[B, C])
```

Figure 5: Python-style pseudocode illustrating model-level optimizations of CEM and autoregressive CBM. Notation for the pseudocode: $B$ is batch size, $C$ is the number of concepts, $d$ is the encoder feature dimension, $E$ is the CEM embedding size, and $H$ is the hidden size. Bias terms are omitted for readability.

avoiding repeated dataset construction and unnecessary CPU–GPU transfers of intermediate states. In this order, the rolling state is moved from CPU to GPU once and then updated directly within the batch loop, without per-step GPU-to-CPU materialization or dataset reconstruction. This optimization explicitly trades bookkeeping storage for runtime. In the step-first implementation, metrics for a given intervention step can be accumulated with a single active accumulator after processing the full test set. In the batch-first implementation, each batch advances through all intervention steps locally, so metric accumulators must be maintained for multiple logged intervention steps within the batch loop. If metrics are logged every $p$ intervention steps, this requires maintaining roughly $\frac{K}{p}$ metric accumulators instead of a single active accumulator. This increases metric-bookkeeping storage, but avoids the dataset reconstruction and host-device transfer costs above and does not add extra metric computation, since the same logged intervention steps are evaluated in both loop orders.

**Intervention evaluation loop order**

**Legacy step-first**

```python
# Full model pass collects intervention tensors.
state_cpu = collect_initial_state(test_loader)
dataset = make_dataset(state_cpu)

# Legacy: outer loop = K steps,
# inner loop = all batches re-scanned K times.
for k in range(K):
    next_state_cpu = []
    for batch in DataLoader(dataset):
        # Re-load materialized intermediate state.
        batch_state = move_to_gpu(batch)

        mask = select_next_concept(batch, k)
        batch_state = apply_intervention(batch_state, mask)
        model.intervene(batch_state)

        # Cost: K CPU<->GPU transfers of rolling state.
        next_state_cpu.append(
            move_to_cpu(batch_state.rolling_state)
        )

    # Cost: K dataset rebuilds from materialized state.
    dataset = make_dataset(concat(next_state_cpu))
```

**Optimized batch-first**

```python
# Full model pass collects intervention tensors once.
state = collect_initial_state(test_loader)

# Optimized: outer loop = batches scanned once,
# inner loop = K steps for the current batch.
for batch_state in iterate_batches(state):
    # Change: one CPU->GPU transfer per batch, not K.
    batch_state = move_to_gpu(batch_state)

    # batch_state stores the intervention mask and
    # model-specific tensors needed by the next step.
    # No dataset rebuild: state advances in place.
    for k in range(K):
        mask = select_next_concept(batch_state, k)
        batch_state = apply_intervention(batch_state, mask)
        model.intervene(batch_state)
```

Figure 6: Python-style pseudocode illustrating the intervention-evaluation loop-order optimization. $K$ is the number of intervention steps.

## D.5 Fast Intervention Solver

The dependency-aware intervention strategy requires solving a constrained optimization problem for the intervention logits. The original implementation uses SLSQP (Kraft, 1988), which repeatedly constructs and solves data-dependent quadratic programming subproblems. Different intervention instances may activate different constraints and require different numbers of internal optimization steps, making these solves difficult to express as a fixed sequence of batched tensor operations on GPUs.

To obtain a GPU-friendly implementation, we instead employ a modified Frank–Wolfe (Frank & Wolfe, 1956) solver. For a constrained optimization problem

$$\min_{\boldsymbol{x}\in\mathbb{D}} f(\boldsymbol{x}),$$

where $\mathbb{D} \subset \mathbb{R}^d$ is a compact convex feasible set, the standard Frank–Wolfe algorithm iteratively solves the linear minimization oracle (LMO)

$$\boldsymbol{y}_t = \arg\min_{\boldsymbol{y}\in\mathbb{D}} \nabla f(\boldsymbol{x}_t)^\top \boldsymbol{y},$$

and updates

$$\boldsymbol{x}_{t+1} = (1 - \gamma_t)\boldsymbol{x}_t + \gamma_t \boldsymbol{y}_t, \qquad \gamma_t \in [0, 1].$$

Because the update is a convex combination, feasibility is preserved whenever $\mathbb{D}$ is convex.

For our intervention problem, the feasible set is

$$\mathbb{D} = \left\{ \boldsymbol{u} : \boldsymbol{u} \geq 0, \ \boldsymbol{u}^\top \boldsymbol{A}\boldsymbol{u} \leq R^2 \right\},$$

where $R^2 = \chi^2_{|\mathbb{S}|,1-\alpha}$ denotes the Mahalanobis-distance threshold introduced in Section 3.2.

The exact LMO therefore becomes

$$\min_{\boldsymbol{u}} \quad \nabla F(\boldsymbol{u}_t)^\top \boldsymbol{u} \quad \text{s.t.} \quad \begin{cases} \boldsymbol{u} \geq \boldsymbol{0}, \\ \boldsymbol{u}^\top \boldsymbol{A}\boldsymbol{u} \leq R^2. \end{cases}$$

The non-negativity constraint makes the exact oracle depend on the unknown active set of coordinates satisfying $u_i = 0$ at the optimum. Different active sets lead to different restricted subproblems, so the exact

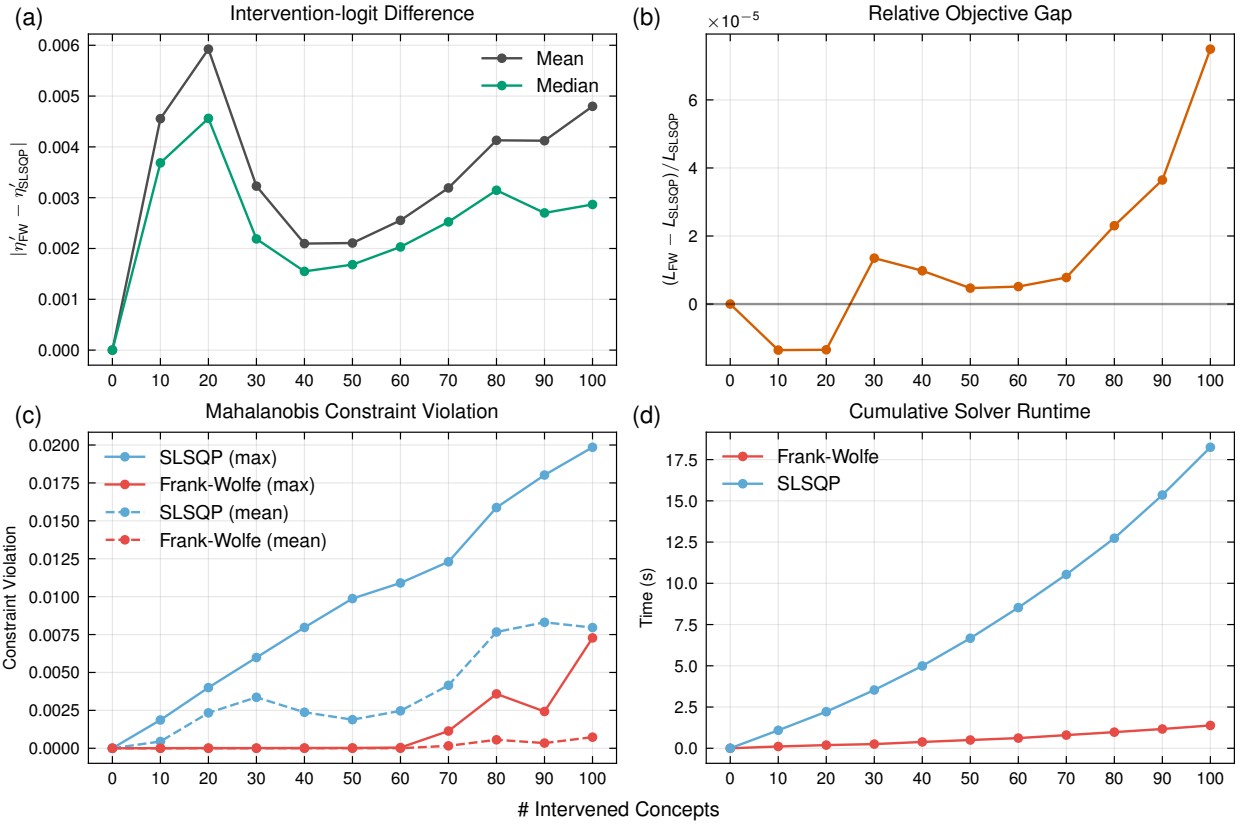

Figure 7: Validation of the modified Frank–Wolfe solver against SLSQP on a batch of 512 samples with 100 concepts. **(a)** Mean and median absolute differences between the intervention logits $\eta'$ produced by the two solvers. **(b)** Relative objective gap $(L_{\text{FW}} - L_{\text{SLSQP}})/|L_{\text{SLSQP}}|$, where positive values indicate that Frank–Wolfe achieves a worse objective than SLSQP. **(c)** Mean and maximum Mahalanobis-distance constraint violations for Frank–Wolfe and SLSQP separately; lower is better. **(d)** Cumulative solver runtime as more concepts are intervened.

oracle does not admit a simple uniform closed-form solution that can be efficiently evaluated across batched intervention instances.

We therefore approximate the oracle by first dropping the non-negativity constraint and solving

$$\min_{\boldsymbol{u}} \quad \nabla F(\boldsymbol{u}_t)^\top \boldsymbol{u} \quad \text{s.t.} \quad \boldsymbol{u}^\top \boldsymbol{A} \boldsymbol{u} \leq R^2.$$

Since the objective is linear, the optimum lies on the ellipsoid boundary unless $\nabla F(\boldsymbol{u}_t) = \boldsymbol{0}$. Introducing a Lagrange multiplier $\lambda \geq 0$, the stationarity condition gives

$$\nabla F(\boldsymbol{u}_t) + 2\lambda \boldsymbol{A} \boldsymbol{u} = 0,$$

and hence

$$\boldsymbol{u}^\star = -\frac{1}{2\lambda} \boldsymbol{A}^{-1} \nabla F(\boldsymbol{u}_t) \propto -\boldsymbol{A}^{-1} \nabla F(\boldsymbol{u}_t).$$

We then approximately enforce $\boldsymbol{u} \geq 0$ by clamping negative coordinates to a small positive value and rescaling the resulting vector back to the ellipsoid boundary. This produces the approximate oracle point $\boldsymbol{v}_t$.

Because this clamp-and-rescale heuristic does not solve the exact LMO, classical Frank–Wolfe convergence guarantees no longer directly apply. To improve robustness, we use a fixed-grid line search on the original

objective:

$$\gamma_t = \arg\min_{\gamma \in \Gamma} F((1 - \gamma)\boldsymbol{u}_t + \gamma \boldsymbol{v}_t),$$

followed by

$$\boldsymbol{u}_{t+1} = (1 - \gamma_t)\boldsymbol{u}_t + \gamma_t \boldsymbol{v}_t.$$

As long as $0 \in \Gamma$, the objective is guaranteed to be non-increasing. Since

$$(1 - 0)\boldsymbol{u}_t + 0 \cdot \boldsymbol{v}_t = \boldsymbol{u}_t,$$

the current iterate itself is included among the candidate solutions considered by the grid search, we have

$$\begin{aligned} F(\boldsymbol{u}_{t+1}) &= F((1 - \gamma_t)\boldsymbol{u}_t + \gamma_t \boldsymbol{v}_t) \\ &= \min_{\gamma \in \Gamma} F((1 - \gamma)\boldsymbol{u}_t + \gamma \boldsymbol{v}_t) \\ &\leq F((1 - 0)\boldsymbol{u}_t + 0 \cdot \boldsymbol{v}_t) \\ &= F(\boldsymbol{u}_t). \end{aligned}$$

Hence,

$$F(\boldsymbol{u}_{t+1}) \leq F(\boldsymbol{u}_t), \qquad \forall t \geq 0.$$

Thus, although our modified solver does not inherit the standard Frank–Wolfe convergence theory, it still preserves amonotonic non-increase of the objective values.

Importantly, the resulting computation consists only of solving batched linear systems involving $\boldsymbol{A}$, clamping, rescaling, and evaluating a fixed set of candidate step sizes. All these operations can be efficiently parallelized on GPUs, making the solver substantially more suitable for large-scale batched intervention evaluation than the original SLSQP-based implementation.

We further compare the proposed solver against the original SLSQP-based strategy on a batch of 512 samples from the synthetic dataset with 100 concepts, in order to assess whether our approximation preserves optimization quality and constraint satisfaction. As shown in Figure 7, the mean absolute difference in intervention logits remains on the order of $10^{-3}$ across all numbers of intervened concepts, indicating that the two solvers produce numerically close solutions. Although the relative objective gap becomes slightly positive at larger intervention sizes, it remains on the order of $10^{-5}$, suggesting comparable optimization quality. For constraint satisfaction, our solver achieves lower mean and maximum Mahalanobis-distance violations than SLSQP across all numbers of intervened concepts, indicating that the constraint is better respected by our solver in this benchmark. Finally, our solver exhibits substantially better scaling with the number of intervened concepts than SLSQP. While the runtime of both solvers increases as more concepts are intervened, the runtime increase is considerably smaller for Frank–Wolfe due to its fixed sequence of batched tensor operations. At 100 intervened concepts, Frank–Wolfe requires only 1.4 s, compared to 18.2 s for SLSQP, corresponding to a 13× speedup. These results suggest that our modified Frank–Wolfe solver with a clamp-and-rescale approximate oracle provides a practical replacement for SLSQP while preserving solution quality.

### D.6 Timing Benchmarks

In Table 4, we compare the runtime of the legacy and optimized implementations. All timings are measured on a single GeForce RTX 4090, thereby isolating the effect of our implementation changes. The benchmarks are reported by dataset, model, and stage, following the same experimental settings described for Table 2.

Table 4: Timing comparison between the legacy and optimized implementations.

| Dataset | Model | Stage | Legacy (s) | Optimized (s) | Speedup |
|---|---|---|---|---|---|
| Synthetic | Autoregressive CBM | pretrain | 49.7 | 2.9 | 17.1× |
| | | concept | 58.9 | 3.6 | 16.4× |
| | | target | 5.6 | 1.6 | 3.5× |
| | | intervention | 576.2 | 45.6 | 12.6× |
| | CEM | joint | 86.9 | 4.7 | 18.5× |
| | | intervention | 62.7 | 37.2 | 1.7× |
| | Hard CBM | joint | 30.2 | 4.8 | 6.3× |
| | | intervention | 63.0 | 38.4 | 1.6× |
| | Amortized SCBM | joint | 16.2 | 11.1 | 1.5× |
| | | intervention | 108.5 | 55.8 | 1.9× |
| | Global SCBM | joint | 14.2 | 9.5 | 1.5× |
| | | intervention | 134.6 | 55.6 | 2.4× |
| CUB | Autoregressive CBM | pretrain | 43.8 | 8.6 | 5.1× |
| | | concept | 52.4 | 13.9 | 3.8× |
| | | target | 20.6 | 1.3 | 15.8× |
| | | intervention | 753.9 | 44.7 | 16.9× |
| | CEM | joint | 67.3 | 15.5 | 4.3× |
| | | intervention | 110.3 | 34.7 | 3.2× |
| | Hard CBM | joint | 36.0 | 15.6 | 2.3× |
| | | intervention | 107.6 | 36.2 | 3.0× |
| | Amortized SCBM | joint | 23.7 | 17.8 | 1.3× |
| | | intervention | 154.3 | 60.9 | 2.5× |
| | Global SCBM | joint | 20.2 | 16.4 | 1.2× |
| | | intervention | 229.0 | 61.8 | 3.7× |
| CIFAR-10 | Autoregressive CBM | pretrain | 188.2 | 8.7 | 21.6× |
| | | concept | 198.4 | 11.4 | 17.4× |
| | | target | 111.8 | 3.6 | 31.1× |
| | | intervention | 1264.6 | 86.1 | 14.7× |
| | CEM | joint | 263.3 | 12.6 | 20.9× |
| | | intervention | 185.5 | 74.5 | 2.5× |
| | Hard CBM | joint | 112.8 | 14.0 | 8.1× |
| | | intervention | 193.3 | 73.3 | 2.6× |
| | Amortized SCBM | joint | 120.2 | 29.3 | 4.1× |
| | | intervention | 278.4 | 104.8 | 2.7× |
| | Global SCBM | joint | 124.6 | 27.5 | 4.5× |
| | | intervention | 504.8 | 104.2 | 4.8× |

