# OpenReview forum: "Towards Practical Reproduction of Stochastic Concept Bottleneck Models"
_TMLR — Under review for TMLR_

### Review · Reviewer_zdPq · 2026-06-26

**Summary Of Contributions:**

In this work the authors attempt to reproduce and improve the practical implementation of Stochastic Concept Bottleneck Models (SCBMs). These models attempt to simultaneously model concepts AND labels associated with datapoints, such that the final label is a function of the concepts generated by the models. This allows for manual concept adjustment to influence labels at test time.

The authors had two goals: to see to what extent the results of previous studies on SCBM were reproducible, and to investigate computational issues in previous studies that led to very expensive computational costs. Their experiments, on both synthetic and natural data, confirmed the findings that SCBMs remain competitive on accuracy metrics, while having good steerability and generally good ECE calibration scores.

On the practical end, the authors improved the data pipeline, vectorization, and Monte Carlo solver to gain a ~ 5x efficiency speedup for SCBM. However these also improved the other baseline methods, often by larger factors, so that in the end SCBMs are not more computationally efficient (as suggested in the original work).

**Audience:**

Yes

**Audience Explanation:**

I am not as familar with the concept bottleneck models research area, but there appears to be enough activity that some readers of TMLR would be interested in papers on the topic. This paper clarifies how SCBMs should be positioned in the literature and gives better implementations for multiple methods in the area. The work

**Claims And Evidence:**

Yes

**Claims Explanation:**

I am not familiar with the area, but the overall experimental setup seems consistent with previous studies. The code optimizations seem sensible and well-validated; the fact that these also speed up the other methods seems to be a robust finding.

**Requested Changes:**

The only thing I would request is that any final version of the paper include the code, as the improved implementations are of the greatest interest to the community.

---

> ### Author Response · Authors · 2026-06-26
> **Response to Reviewer zdPq**
>
> Dear Reviewer zdPq,
>
> Thank you very much for your encouraging review. We are pleased that you found our experimental evidence convincing and that you consider our implementation improvements valuable to the community.
>
> We also appreciate your suggestion regarding code availability. Our implementation is already included in the Supplementary Material for the review process. In the final version, we will additionally provide a public GitHub repository link to make the code easily accessible and facilitate reproducibility.
>
> Thank you again for your helpful feedback.

---

> ### Author Response · Authors · 2026-07-18
> **Revision uploaded**
>
> Dear Reviewer zdPq,
>
> We have uploaded a revised version of the manuscript that incorporates clarifications requested by the other reviewers. Regarding your suggestion, we confirm that the code will be made available via a public GitHub repository in the final version. Please let us know if you have any further questions.
>
> Best regards,
>
> The Authors

---

### Review · Reviewer_CpMu · 2026-07-03

**Summary Of Contributions:**

The paper studies the Stochastic Concept Bottleneck Model (SCBM), and reproduce a subset of the experiments from the original paper that first proposed the SCBM models, i.e. Vandenhirtz et al. (2024).

The paper finds that the main conclusions are reproducible and they observe that the practical implementation from Vandenhirtz et al. (2024) can be improved and provide a high-level description of the improvements.

**Additional Comments:**

- The paper says: "To mitigate information leakage, SCBMs follow the approach".... What information leakage exactly. This should be elaborated.
- Fig. 1 could be improved by making the encoder block output u(x) and sigma(x) directly rather than f(x)

**Audience:**

No

**Audience Explanation:**

I think there is value in reproduction studies, even though it is not tradition in ML. However, I also believe that the introduction and motivation are somewhat confusing and not specific, and therefore, I think the TMLR audience will have a limited interest in the paper in its current form.

**Claims And Evidence:**

No

**Claims Explanation:**

The main claim is that  the empirical conclusion from Vandenhirtz et al. (2024) are reproducible. However, the paper is not very specific about what experiments and conclusions they are trying to replicate. Therefore, it is somewhat hard to assess whether the claims are supported in the current version of the paper.

**Requested Changes:**

- The introduction says: "The paper explains that the authors profiled the official implementation, made some observations on low and unstable GPU utilization, and then continues: "This naturally lead us to investigate whether the main empirical conclusions of SCBM hold...". This link is not clear to me and should be elaborated. Sure, claims about relative run-time may be affected, but claims about predictive accuracy etc. and not affected by inefficient implementation, and therefore, I am not able to follow the motivation for the investigation, Moreover, the authors should state exactly what claims in the original paper that they are referring to.

- The paper states: ".... we reproduce the main SCBM experiments on a synthetic dataset and two natural image datasets.". This formulation makes it unclear whether the current paper is aiming to reproduce the experiments from Vandenhirtz et al, (2024) exactly, or whether they run the same experimental design, but for different datasets. This should be clarified.

- The authors mention: "We show that runtime comparisons are sensitive to implementation choices" as a contribution, but this is in general quite well-known and well-documented already, so it is hard for me to see the contribution here in the current version of the paer. Therefore, the authors should strive to motivate this better.

- The paper states "An interesting pattern can be observed in the CIFAR-10 dataset. Although CEM has the lowest concept accuracy among the compared models, it achieves the highest target accuracy.". Maybe I am missing something, but it does not really seem like a "pattern" to me if the phenomenon only happens in 1 out of 3 datasets. The authors should clarify or rephrase this.

- The paper states: "Beyond reproducing the empirical results, we study the practical cost of reproducing SCBM". Why is it interesting to study the practical cost of reproduction? The authors should elaborate on this.

- In sec 6. the paper superficially describes some changes to the code which lead to a speed up, and appendix D elaborates a bit further. However, if the authors really contributed with some non-trivial and clever implementation details, these should be explained in much greater detail for the field appreciate them.

---

> ### Author Response · Authors · 2026-07-04
> **Response to Reviewer CpMu**
>
> Dear Reviewer CpMu,
>
> Thank you for the careful reading and detailed feedback.
>
> We agree with the reviewer’s underlying concern. Because we conducted this reproduction study ourselves, we are familiar with Vandenhirtz et al. (2024) and its experimental details, but we should not have assumed that readers share this background. Upon reflection, several parts of our manuscript relied too much on context from the original paper and related work without making that context explicit. This made our motivation, experimental scope, and contributions harder to follow than they should be.
>
> We appreciate the reviewer for pointing this out. In the revised manuscript, we will make the paper more self-contained by adding the necessary context throughout. In particular, we will clarify:
>
> 1. the specific claims in Vandenhirtz et al. (2024) that motivate our reproduction study;
> 2. the exact scope of what we reproduce versus what we extend or additionally analyze;
> 3. technical terms, such as “information leakage,” that were previously not explained sufficiently;
> 4. the implementation changes behind the reported speedups, especially where the changes are non-trivial and relevant to our contribution.
>
> Our goal is to ensure that readers can understand the motivation and contributions of our paper without needing to have read Vandenhirtz et al. (2024) in advance.
>
> Below we outline how we address each point.

---

> ### Author Response · Authors · 2026-07-04
> **Response to Requested Changes**
>
> ### 1. Motivation link between profiling and empirical reproduction
>
> **Reviewer concern.**
> The paper currently makes the profiling analysis sound like the motivation for reproducing the empirical results, which creates a confusing causal link.
>
> **Response.**
> Our primary goal from the outset was to reproduce the main empirical claims of Vandenhirtz et al. (2024). The practical-efficiency analysis was separate from this empirical reproduction goal. It arose because the original paper estimates that reproducing the full experimental suite requires approximately 1,200 GPU hours on GeForce RTX 2080 hardware, and because the released implementation remained costly enough to make full reproduction nontrivial even on a more recent GeForce RTX 4090.
>
> We agree that the original phrasing, "This naturally led us to investigate whether the main empirical conclusions of SCBM hold...", incorrectly implied that the efficiency issue motivated the empirical reproduction. This was not our intention.
>
> **Change made.**
> We revise the introduction to separate the empirical reproduction goal from the practical reproduction-cost analysis. We also explicitly list the main SCBM claims we reproduce, so the motivation for the empirical study no longer depends on the implementation-efficiency observations.
>
> ---
>
> ### 2. Scope of reproduced experiments and datasets
>
> **Reviewer concern.**
> The phrase "we reproduce the main SCBM experiments on a synthetic dataset and two natural image datasets" is ambiguous. It is unclear whether the experiments exactly reproduce the original datasets and settings, or apply the same experimental design to different datasets.
>
> **Response.**
> We agree that the original wording was ambiguous.
>
> **Change made.**
> We clarify the reproduction scope in both the introduction and the “Datasets and Evaluation” section. The revised text states that we reproduce the main-text experiments from Vandenhirtz et al. (2024), use the same natural-image datasets and concept-label sources, and use the same synthetic data-generation procedure with adjusted hyperparameters that are documented in the appendix.
>
> Because the reviewer's quoted sentence appears in the abstract, we keep the abstract concise while adding the full dataset-by-dataset clarification in the introduction and main experimental setup.
>
> ---
>
> ### 3. Contribution of runtime sensitivity to implementation choices
>
> **Reviewer concern.**
> The observation that runtime comparisons are sensitive to implementation choices is already well known, so the contribution needs stronger motivation.
>
> **Response.**
> We agree that the general statement is not novel. However, our intended contribution was more specific: we revisit the runtime-related claim made in Vandenhirtz et al. (2024). The original paper presents SCBM as retaining the training and inference efficiency of CBMs and compares its runtime favorably against Autoregressive CBM. After optimizing avoidable implementation overhead, we find that the optimized implementation does not support the original conclusion that SCBM is more efficient than Autoregressive CBM.
>
> Thus, our result should be interpreted as a qualification of the original SCBM runtime claim, not as a broad claim about runtime benchmarking in general.
>
> **Change made.**
> We revise the abstract, contribution statement, and runtime-results discussion to frame this as a re-evaluation of the original SCBM runtime comparison, rather than as a general observation about runtime benchmarking. In the "Practical Reproduction Findings" section, we now explicitly state that our optimized implementation does not support the original conclusion that SCBM is more efficient than Autoregressive CBM, and suggest that SCBM's practical advantage is better supported by its ability to exploit dependencies among concepts during intervention than by raw computational efficiency.
>
> ---

---

> ### Author Response · Authors · 2026-07-04
> **Response to Requested Changes (Continued)**
>
> ### 4. CIFAR-10 concept accuracy versus target accuracy observation
>
> **Reviewer concern.**
> The manuscript describes this as a "pattern," but the phenomenon appears in only one of the three reported datasets.
>
> **Response.**
> We agree that the word "pattern" was too strong. This observation appears only on CIFAR-10 among the datasets we report, and we did not intend to claim that it is consistent across all datasets.
>
> Our reason for highlighting this result was that it relates to a broader issue discussed in the concept-bottleneck literature: information leakage and the expressiveness-faithfulness trade-off. CEM passes high-dimensional concept embeddings to the target predictor, so target accuracy can improve even when scalar concept accuracy is lower if the embeddings carry task-relevant information not captured by the binary concept values. We did not explain this background clearly enough in the original manuscript.
>
> **Change made.**
> We revise the wording to present this as a CIFAR-10-specific discrepancy rather than a general pattern. We also add background on information leakage in the Related Work section, explain why this issue is relevant for CEM's high-dimensional embeddings, and connect the CIFAR-10 observation to the sufficiency of the concept bottleneck in the Main Results section.
>
> This should make clear why the observation is relevant without overstating its generality.
>
> ---
>
> ### 5. Motivation for studying practical reproduction cost
>
> **Reviewer concern.**
> The paper should better explain why studying the practical cost of reproduction is interesting.
>
> **Response.**
> We agree. As discussed above, practical reproduction cost was not the original motivation for our empirical reproduction study. It became important because the original paper estimates that reproducing the full experimental suite requires approximately 1,200 GPU hours on GeForce RTX 2080 hardware. Such a cost makes full reproduction difficult in practice for other researchers and uses compute that could otherwise support additional baselines, ablations, or robustness checks.
>
> Efficient implementations matter for practical reproducibility. In our setting, experiments are run on rented cloud GPUs, so reducing runtime directly lowers the monetary cost of verification. Even when monetary cost is not the primary constraint, faster reproduction enables researchers to run additional baselines, ablations, and robustness checks under the same time and compute budget.
>
> **Change made.**
> We add this motivation to the introduction and present the cost analysis explicitly as a secondary part of the study that supports practical reproducibility.
>
> ---
>
> ### 6. Detail of implementation changes and speedup
>
> **Reviewer concern.**
> Section 6 and Appendix D describe the code changes too superficially. If the implementation details are part of the contribution, they should be explained in greater detail.
>
> **Response.**
> We agree that the current manuscript does not provide enough detail for some of the changes behind the speedups. We now distinguish between two types of changes. First, several changes are engineering optimizations that preserve the original computation but reduce avoidable overhead. We keep these details in the appendix, but add more concrete explanations and pseudocode for the less obvious cases so that readers can understand what changed structurally. Second, the fast intervention solver is an algorithmic optimization of the dependency-aware intervention procedure. Since this solver changes how the constrained intervention problem is solved, we describe it in the Method section and provide additional details and validation in the appendix.
>
> **Change made.**
> For the engineering optimizations, we expand the appendix and add Python-style pseudocode for the CEM optimization, the autoregressive CBM optimization, and the intervention-evaluation loop-order optimization. For the algorithmic optimization, we add a Method subsection describing the batched Frank-Wolfe intervention solver, including the reparameterization, approximate oracle, clamp-and-rescale step, and grid line search; the appendix further reviews its relation to classical Frank-Wolfe guarantees and validates it empirically against SLSQP.
>
> ---
>
> ## Response to Additional Comments
>
> ### 7. Information leakage in SCBM
>
> **Reviewer concern.**
> The phrase "To mitigate information leakage, SCBMs follow the approach..." does not explain what information leakage means.
>
> **Response.**
> Please check our response to Point 4.
>
> ---
>
> ### 8. Figure 1 encoder output notation
>
> **Reviewer concern.**
> Figure 1 could be improved by making the encoder block output `u(x)` and `sigma(x)` directly rather than `f(x)`.
>
> **Response.**
> We agree that Figure 1 can be made clearer. In the revised manuscript, we modify the encoder block so that it directly outputs $\mu(x)$ and $\Sigma(x)$ instead of the less informative notation $f(x)$.

---

> ### Author Response · Authors · 2026-07-18
> **Revision uploaded**
>
> Dear Reviewer CpMu,
>
> We have uploaded a revised version of the manuscript incorporating the changes discussed in our responses above. Please let us know if you have any further questions or concerns.
>
> Best regards,
> The Authors

---

### Review · Reviewer_a1sK · 2026-07-18

**Summary Of Contributions:**

The paper reproduces results from Stochastic Concept Bottleneck Models. The paper reaffirms the findings of the original paper and proposes implementation improvements that provide around 4-5x speedup.

**Audience:**

No

**Audience Explanation:**

I answered the above question negatively based on the reasoning in an example used on TMLR's Acceptance Criteria page, which I reproduce below.

"A machine learning class report that re-runs the experiments of a published paper has educational value to the students involved. But if it doesn't surface generalizable insights, it is unlikely to be of interest to (even a subset of) the TMLR audience, and so could be rejected based on this criterion."

Although the submission is of higher quality than a class report, I don't know what insights it provides to a general reader of ML literature. I feel that the only improvement that this submission has over SCBMs is the improved implementation, which, by itself, in my view, does not warrant a paper; it could be a GitHub repository. But I am open to arguments against this view from the authors, other reviewers, and AE.

**Claims And Evidence:**

Yes

**Claims Explanation:**

I understood the claim to be that SCBMs are a useful/competitive class of CBMs. This claim is verified by the paper. But I am not sure if this is a contribution.

**Requested Changes:**

My primary concern is that there isn't anything significant to learn from this paper. The submission has fewer experiments on the core idea of SCBMs than the original paper (see its appendix). But it does have more experiments on the implementation side. Can the authors obtain more insights into SCBMs through their reproducibility study?

Some suggestions (insights do not have to be related to these): (1) comparing SCBMs against baselines on their ability to extract insights from pre-trained models/representations, (2) learning dynamics of SCBMs, (3) causal agreement between concepts in contrast to simple correlation between concepts, (4) rare concept (e.g., heavy-tailed distributions) identification.

---

> ### Author Response · Authors · 2026-07-18
> **Response to reviewer a1sK**
>
> Dear reviewer a1sK,
>
> We thank the reviewer for raising the important question of what the TMLR community can learn from our work. This review made us realize that the original manuscript did not sufficiently emphasize the broader insights and practical lessons obtained from our reproduction study, which may have made the paper appear closer to a simple rerun than intended. To address this issue, **we revised the abstract, introduction, and contribution statements to make these insights more explicit**.
>
> At the same time, we would like to emphasize that our work is not limited to rerunning the original experiments. Besides independently reproducing the main empirical claims, we systematically study the practical cost of reproducing SCBM and identify major implementation bottlenecks in the released codebase. In the revised manuscript we added a detailed description of the algorithmic modifications introduced in our reproduction pipeline. Specifically, **we replace the original per-sample CPU-based constrained optimization procedure with a batched GPU-compatible Frank-Wolfe solver**. We believe that this goes beyond a simple rerun and instead reflects an algorithm-system co-design perspective. Rather than simply accelerating the original implementation, we reformulate the optimization procedure into a form that is more amenable to GPU while preserving the original optimization objective and intervention semantics.
>
> Beyond these algorithmic modifications, we also include more engineering details and optimizations in the appendix that accelerate SCBM and other CBM variants. We believe that these practical details may be useful for future researchers who wish to reproduce, extend, or benchmark these methods.
>
> Importantly, despite all these implementation changes, the main qualitative conclusions of the original paper remain unchanged. We believe this itself constitutes an important finding: **the core empirical advantages of SCBM are robust to changes in implementation details**. At the same time, our study also reveals that not all conclusions made by the original paper exhibit the same robustness. After removing avoidable implementation overheads, **the relative runtime behavior of the compared methods changes**. This suggests that the computational-efficiency claims made in the original paper are implementation-dependent and should be interpreted with caution. Consequently, **our results indicate that the practical value of SCBM lies primarily in dependency-aware interventions rather than computational-efficiency advantages**. We believe this provides useful guidance for future researchers regarding how SCBM should be positioned among existing CBM variants and when it constitutes an appropriate baseline.

---

> ### Author Response · Authors · 2026-07-18
> **Response to reviewer a1sK (continued)**
>
> We also appreciate the reviewer's suggestions regarding additional possible analyses. However, we believe that these suggestions correspond to adjacent but different research questions from those studied by SCBM.
>
> (1) *Comparing SCBMs based on their ability to extract insights from pretrained representations would require a different problem formulation*. In the SCBM setting, the concept vocabulary and concept annotations are predefined. SCBM is not designed for concept discovery or representation probing, and all compared methods are fundamentally limited to the provided concept space.
>
> (2) Regarding *learning dynamics*, SCBM is not motivated by specific optimization dynamics considerations. Without a concrete hypothesis, simply presenting training curves would likely provide limited additional insight. The primary motivation of SCBM concerns intervention behavior, and our reproduction already evaluates how target predictions evolve under successive concept corrections.
>
> (3) Studying *causal agreement between concepts* would go beyond the assumptions of SCBM. The covariance structure learned by SCBM models statistical dependencies rather than causal relationships. Investigating concept-level causality would require additional assumptions or datasets with causal ground truth and would effectively constitute a new research direction in causal concept modeling.
>
> (4) Similarly, *rare-concept identification* is not part of the SCBM problem setting. Concepts and their labels are assumed to be given, and SCBM does not aim to discover concepts from data.

---

> ### Author Response · Authors · 2026-07-18
> **Response to reviewer a1sK (continued)**
>
> Overall, we believe the main insights from our work can be summarized as follows:
>
> (i) **the principal empirical findings of SCBM can be independently reproduced and remain robust under implementation and algorithmic changes**;
>
> (ii) the reported computational advantage of SCBM over autoregressive CBMs is considerably more sensitive to implementation choices than suggested by the original paper. Consequently, **dependency-aware interventions, rather than computational efficiency, appear to constitute the more robust practical advantage of SCBM.** It is important for future researchers to position SCBM appropriately among existing CBM variants;
>
> (iii) **reducing reproduction cost is itself valuable for the community**, as it enables future researchers to evaluate additional baselines, perform more extensive ablations, and investigate robustness questions in less time.
>
> To better communicate these insights, we revised the abstract, introduction, contribution statements, and conclusion in the revised manuscript.
>
> For these reasons, we believe that our work is more closely aligned with the positive example described in the TMLR acceptance criteria: **a reproducibility study that systematically investigates robustness and provides actionable lessons for future research, rather than a simple rerun of previously published experiments.** We hope that these revisions and clarifications make the intended contributions and practical lessons of our work more explicit.

---

> ### Author Response · Authors · 2026-07-18
> **Revision uploaded**
>
> Dear Reviewer a1sK,
>
> We have uploaded a revised version of the manuscript that incorporates the changes discussed in our responses above. In particular:
>
> - Abstract and Introduction: revised to explicitly state the  main insights summarized in our response above.
>
> - Method section: added a new subsection `3.3 Solver for Dependency-Aware Interventions` describing the batched Frank-Wolfe solver we introduce to replace the original per-sample CPU-based method, highlighting this as a methodological contribution instead of a pure engineering speedup.
>
> - Appendix: expanded with pseudocode and additional detail for the engineering optimizations, and added the derivation showing how the solver was designed to make this optimization amenable to batched GPU computation, a proof that the modified update preserves monotonic descent
>
> - Conclusion: revised to explicitly state our three main takeaways — that SCBM's core empirical findings are robust to substantial implementation and algorithmic changes, that its reported computational advantage is considerably more implementation-dependent than these other findings, and that lowering the reproduction cost itself benefits the community by making it feasible to run additional baselines and ablations under the same compute budget.
>
> We hope these revisions make the contributions and practical insights of our work clearer, and we would be happy to address any remaining concerns.
>
> Best regards,
>
> The Authors

---

> > ### Comment · Reviewer_a1sK · 2026-07-18
> > **Reviewer a1sK's response to the authors**
> >
> > I thank the authors for their detailed response and for updating the PDF. Unfortunately, I still stand by my original evaluation. While implementation improvements are valuable, I am not sure if they are sufficient to warrant a paper, especially given that there are no new insights. As the TMLR sentences that I quoted say, without generalizable insights, submissions cannot be considered valuable to the community/audience. The implementation improvements mentioned in this work could be a GitHub repo or a Python library.
> >
> > A similar observation was made by reviewer CpMu: "...claims about relative run-time may be affected, but claims about predictive accuracy etc. and not affected by inefficient implementation..." I have read the authors' response to their concerns. But I agree with reviewer CpMu that the claim in the original paper did not depend on its runtime efficiency.

---

> > > ### Author Response · Authors · 2026-07-18
> > > **Response to Reviewer a1sK**
> > >
> > > Dear Reviewer a1sK,
> > >
> > > Thank you for engaging further with our response. We would like to respond to two specific points raised above.
> > >
> > > ---
> > >
> > > ## On generalizability of the insight
> > > The authors of the original SCBM paper explicitly attribute the slow sampling process of Autoregressive CBM to its autoregressive structure, and state that SCBM retains the efficiency of standard CBMs by contrast. Concretely, the original paper reports approximately a **$15 \times$ difference in inference time** between SCBM and Autoregressive CBM. This is presented as one of the main empirical claims of the paper, alongside predictive/calibration performance.
> > >
> > > In our manuscript, we show that once avoidable implementation overhead is removed, the relative runtime behavior of the compared models changes substantially: **Autoregressive CBM becomes even faster than SCBM**. This directly **overturns one of the main empirical claims in the original paper**. We believe this constitutes a concrete, actionable insight for future researchers: SCBM should not be adopted or cited for its computational efficiency over Autoregressive CBM; its practical value lies specifically in dependency-aware intervention.
> > >
> > > We respectfully disagree that a finding which reverses a headline empirical claim of the original paper should be characterized as *offering no insight*.
> > >
> > > ---
> > >
> > > ## On Reviewer CpMu's comment
> > >
> > > We would like to clarify the context of the comment quoted above. Reviewer CpMu's original concern (Point 1 in our detailed response) was about the motivation connecting our profiling analysis to our empirical reproduction — specifically, that our introduction implied the efficiency issue motivated us to question SCBM's predictive-accuracy claims, which would indeed be an unjustified causal link, since accuracy claims are not affected by implementation inefficiency.
> > >
> > > We agree with this concern and have revised the introduction accordingly to separate these two goals: our empirical reproduction was motivated independently, and the efficiency analysis is a separate contribution in its own right.
> > >
> > > Reviewer CpMu's comment was made in the context of critiquing how our introduction framed the motivation for our reproduction study, not as a judgment on whether claims in the original paper depend on implementation. We therefore do not think this comment can be read as supporting the view that the original paper's claims — including, specifically, the runtime-efficiency claim — are independent of implementation choices.
> > >
> > > We hope this clarifies both the nature of our contribution and the context of Reviewer CpMu's original comment. We remain happy to discuss further.

---

> > > > ### Comment · Reviewer_a1sK · 2026-07-18
> > > > **Reviewer a1sK's response**
> > > >
> > > > I once again thank the authors for their effort in the rebuttal. I accept their point about reviewer CpMu's comment. However, I still disagree with the authors on my primary contention.
> > > >
> > > > This is the only sentence I found in the SCBM paper that compared SCBM's speed with Autoregressive CBM's (beginning of page 8 in their current ArXiv version):
> > > >
> > > > > It is evident that the autoregressive CBM of Havasi et al. (2022) suffers from a slow sampling process due to its autoregressive structure, while SCBMs retain the efficiency of CBMs.
> > > >
> > > > Through the experiments in the present submission, the authors have exonerated Autoregressive CBMs. But the SCBM paper's claim that SCBM retained the efficiency of CBMs still stands. In other words, the results on Autoregressive CBMs do not overturn any empirical claims from the original work.
> > > >
> > > > Now, one may compare SCBMs with CBMs in Table 2 of the submission to refute the claim that "SCBMs maintain the efficiency of CBMs." I don't know the source of this difference. But even if this improvement was due to intentional changes in their implementation, it would still be a minor claim, since computational efficiency was not the primary motivation behind SCBMs.

---

> > > > > ### Author Response · Authors · 2026-07-18
> > > > > **Response to Reviewer a1sK**
> > > > >
> > > > > Dear Reviewer a1sK,
> > > > >
> > > > > Thank you again for the detailed follow-up and for pointing us to the exact sentence you have in mind. Indeed, the sentence you quoted
> > > > > ```text
> > > > > In Table 2, we show the time it takes for training and testing of the methods. It is evident that the autoregressive CBM of Havasi et al. (2022) suffers from a slow sampling process due to its autoregressive structure, while SCBMs retain the efficiency of CBMs.
> > > > > ```
> > > > > is the only sentence we found in the SCBM paper that directly compares SCBM's speed with Autoregressive CBM's.
> > > > >
> > > > > We do not think this sentence should be read in isolation, without considering its context. The sentence contrasts two things in one breath, following the structure "X suffers from slowness... while Y retains the efficiency of Z" (where X = Autoregressive CBM, Y = SCBM, Z = standard CBM). This structure puts Autoregressive CBM and SCBM side by side on purpose — **both model dependencies among concepts**, and the sentence is comparing how each one affects speed, with Z (standard CBM) serving as the reference point for what "efficient" means.
> > > > >
> > > > > If the intended comparison were purely SCBM versus CBM and unrelated to Autoregressive CBM, this contrastive framing would not be the natural way to phrase it.  We therefore believe it is reasonable to read this sentence as making an implicit comparative claim: SCBM is more efficient than Autoregressive CBM.
> > > > >
> > > > > This reading is reinforced by how the efficiency claim is framed elsewhere in the paper. The abstract says their proposed method "allows SCBMs to retain the CBMs' efficient training and inference procedure," and the conclusion says the same thing again: "our modeling approach retains CBMs' training and inference speed." This shows that the efficiency claim is **repeatedly emphasized throughout the paper** rather than being mentioned only in a single experimental result. But we *don't* think the fact that the text only names "CBM" means the claim is limited to a comparison with standard CBM.
> > > > >
> > > > > Standard CBM is chosen as the reference point for "efficient" precisely because it does not model concept correlations — so matching its speed shows that SCBM remains efficient even though it models correlations. Autoregressive CBM is the **only other baseline that also models concept correlations**, so it has to be part of the intended scope of this efficiency claim: it is the natural test of whether a correlation-modeling method can still match CBM's speed, and Table 2 shows that Autoregressive CBM fails to do so. The efficiency discussion in Table 2 therefore naturally invites a
> > > > > comparison between SCBM and Autoregressive CBM, since both methods
> > > > > explicitly model concept dependencies while standard CBM serves as
> > > > > the efficiency reference point.
> > > > >
> > > > > For these reasons, we believe that our findings lead to a refined interpretation of the practical advantages of SCBM. We believe that this distinction provides actionable guidance for
> > > > > future researchers when positioning SCBM among existing CBM variants.
> > > > >
> > > > > We are happy to discuss further if the reviewer has any remaining concerns.
> > > > >
> > > > >
> > > > > Best regrads,
> > > > >
> > > > > The Authors

---

> > > > > ### Author Response · Authors · 2026-07-19
> > > > > **Response to Reviewer a1sK (continued)**
> > > > >
> > > > > Dear Reviewer a1sK,
> > > > >
> > > > > We would like to add one point that we think sharpens the argument above.
> > > > >
> > > > > The original paper's efficiency claim is really answering a question: *does modeling concept correlations slow the model down?* Their answer is no. SCBM is as fast as standard CBM, while Autoregressive CBM, the only other baseline that also models concept correlations, is not. The comparison to Autoregressive CBM is what shows this is a real achievement: it demonstrates that modeling concept correlations does not have to come at a speed cost, precisely because the other method (Autoregressive CBM) that tries to model correlations does pay that cost.
> > > > >
> > > > > However, once implementation overhead is removed, this contrast goes away. Autoregressive CBM matches, and even beats, SCBM's speed. So the claim still holds in the sense that SCBM does capture concept dependencies without paying a speed cost. What no longer holds is the *implicit claim that this is something only SCBM can do; Autoregressive CBM can achieve the same thing*.
> > > > >
> > > > > As we argued before, **the efficiency claim is not a minor claim**; it appears prominently in the abstract and conclusion of the original SCBM paper as an advantage. Our findings indicate that this particular claim is substantially more implementation-dependent than the other main findings. By clarifying that the speed advantage over Autoregressive CBM largely disappears under a fair comparison, **our work refines the understanding of where SCBM’s practical value truly lies**. We believe pointing this out is a useful, generalizable insight from our reproduction study, **helping future researchers position SCBM among existing CBM variants and choose appropriate baselines for their own work**.
> > > > >
> > > > > We are happy to discuss further if the reviewer has any remaining concerns.
> > > > >
> > > > > Best regards,
> > > > >
> > > > > The Authors